# MUTUAL CALIBRATION BETWEEN EXPLICIT AND IMPLICIT DEEP GENERATIVE MODELS

## ABSTRACT

Deep generative models are generally categorized into explicit models and implicit models. The former defines an explicit density form that allows likelihood inference; while the latter targets a flexible transformation from random noise to generated samples. To take full advantages of both models, we propose *Stein Bridging*, a novel joint training framework that connects an explicit (unnormalized) density estimator and an implicit sample generator via Stein discrepancy. We show that the Stein bridge 1) induces novel mutual regularization via kernel Sobolev norm penalization and Moreau-Yosida regularization, and 2) stabilizes the training dynamics. Empirically, we demonstrate that Stein Bridging can facilitate the density estimator to accurately identify data modes and guide the sample generator to output more high-quality samples especially when the training samples are contaminated or limited.

## 1 INTRODUCTION

Deep generative model, as a powerful unsupervised framework for learning the distribution of high-dimensional multi-modal data, has been extensively studied in recent literature. Typically, there are two types of generative models: explicit and implicit (Goodfellow et al., 2014). Explicit models define a density function of the distribution, while implicit models learn a mapping that generates samples by transforming an easy-to-sample random variable.

Both models have their own power and limitations. The density form in explicit models endows them with convenience to characterize data distribution and infer the sample likelihood. However, the unknown normalizing constant often causes computational intractability. On the other hand, implicit models including generative adversarial networks (GANs) can directly generate vivid samples in various application domains including images, natural languages, graphs, etc. (Goodfellow et al., 2014; Radford et al., 2016; Arjovsky et al., 2017; Brock et al., 2019). Nevertheless, one important challenge is to design a training algorithm that do not suffer from instability and mode collapse. In view of this, it is natural to build a unified framework that takes full advantages of the two models and encourages them to compensate for each other.

Intuitively, an explicit density estimator and a flexible implicit sampler could help each other's training given effective information sharing. On the one hand, the density estimation given by explicit models can be a good metric that measures quality of samples (Dai et al., 2017), and thus can be used for scoring generated samples given by implicit model or detecting outliers as well as noises in input true samples (Zhai et al., 2016). On the other hand, the generated samples from implicit models could augment the dataset and help to alleviate mode collapse especially when true samples are insufficient that would possibly make explicit model fail to capture an accurate distribution. We refer to Appendix A for a more comprehensive literature review.

Motivated by the discussions above, in this paper, we propose a joint learning framework that enables mutual calibration between explicit and implicit generative models. In our framework, an explicit model is used to estimate the unnormalized density; in the meantime, an implicit generator model is exploited to minimize certain statistical distance (such as the Wasserstein metric or Jensen-Shannon divergence) between the distributions of the true and the generated samples. On top of these two models, a Stein discrepancy, acting as a *bridge* between generated samples and estimated densities, is introduced to push the two models to achieve a consensus. Unlike flow-based models (Nguyen et al., 2017; Kingma & Dhariwal, 2018; Papamakarios et al., 2017), our formulation does not impose

invertibility constraints on the generative models and thus is flexible in utilizing general neural network architectures. Our main contribution are as follows.

- Theoretically, we prove that our method allows the two generative models to impose novel mutual regularization on each other. Specifically, our formulation penalizes large kernel Sobolev norm of the critic in the implicit (WGAN) model, which ensures the critic not to change suddenly on the high-density regions and thus preventing the critic of the implicit model being to strong during training. In the mean time, our formulation also smooths the function given by the Stein discrepancy through Moreau-Yosida regularization, which encourages the explicit model to seek more modes in the data distribution and thus alleviates mode collapse.

- In addition, we also show that the joint training helps to stabilize the training dynamics. Compared with other common regularization approaches for GAN models that may shift original optimum, our method can facilitate convergence to unbiased model distribution.

- Extensive experiments on synthetic and image datasets justify our theoretical findings and demonstrate that joint training can help two models achieve better performance. On the one hand, the energy model can detect complicated modes in data more accurately and distinguish out-of-distribution samples. On the other hand, the implicit model can generate higher-quality samples, especially when the training samples are contaminated or limited.

## 2 BACKGROUND

We briefly provide some technical background related to our model.

**Energy Model.** The energy model assigns each data $\mathbf{x} \in \mathbb{R}^d$ with a scalar energy value $E_\phi(\mathbf{x})$, where $E_\phi(\cdot)$ is called energy function and is parameterized by $\phi$. The model is expected to assign low energy to true samples according to a Gibbs distribution $p_\phi(\mathbf{x}) = \exp\{-E_\phi(\mathbf{x})\}/Z_\phi$, where $Z_\phi$ is a normalizing constant dependent of $\phi$. The normalizing term $Z_\phi$ is often hard to compute, making the training intractable, and various methods are proposed to detour such term (see Appendix A).

**Stein Discrepancy.** Stein discrepancy (Gorham & Mackey, 2015; Liu et al., 2016; Chwialkowski et al., 2016; Oates et al., 2017; Grathwohl et al., 2020) is a measure of closeness between two probability distributions and does not require knowledge for the normalizing constant of one of the compared distributions. Let $\mathbb{P}$ and $\mathbb{Q}$ be two probability distributions on $\mathcal{X} \subset \mathbb{R}^d$, and assume $\mathbb{Q}$ has a (unnormalized) density $q$. The Stein discrepancy $\mathcal{S}(\mathbb{P}, \mathbb{Q})$ is defined as

$$\mathcal{S}(\mathbb{P}, \mathbb{Q}) := \sup_{\mathbf{f} \in \mathcal{F}} \mathbb{E}_{\mathbf{x} \sim \mathbb{P}}[\mathcal{A}_{\mathbb{Q}} \mathbf{f}(\mathbf{x})] := \sup_{\mathbf{f} \in \mathcal{F}} \{\Gamma(\mathbb{E}_{\mathbf{x} \sim \mathbb{P}}[\nabla_{\mathbf{x}} \log q(\mathbf{x}) \mathbf{f}(\mathbf{x})^\top + \nabla_{\mathbf{x}} \mathbf{f}(\mathbf{x})])\}, \quad (1)$$

where $\mathcal{F}$ is often chosen to be a Stein class (see, e.g., Definition 2.1 in (Liu et al., 2016)), $\mathbf{f} : \mathbb{R}^d \to \mathbb{R}^{d'}$ is a vector-valued function called *Stein critic* and $\Gamma$ is an operator that transforms a $d \times d'$ matrix into a scalar value. One common choice of $\Gamma$ is trace operator when $d' = d$. One can also use other forms for $\Gamma$, like matrix norm when $d' \neq d$ (Liu et al., 2016). If $\mathcal{F}$ is a unit ball in some reproducing kernel Hilbert space (RKHS) with a positive definite kernel $k$, it induces Kernel Stein Discrepancy (KSD). More details are provided in Appendix B.

**Wasserstein Metric.** Wasserstein metric is suitable for measuring distances between two distributions with non-overlapping supports (Arjovsky et al., 2017). The Wasserstein-1 metric between distributions $\mathbb{P}$ and $\mathbb{Q}$ is defined as

$$\mathcal{W}(\mathbb{P}, \mathbb{Q}) := \min_{\gamma} \mathbb{E}_{(\mathbf{x},\mathbf{y}) \sim \gamma}[\|\mathbf{x} - \mathbf{y}\|],$$

where the minimization with respect to $\gamma$ is over all joint distributions with marginals $\mathbb{P}$ and $\mathbb{Q}$. By Kantorovich-Rubinstein duality, $\mathcal{W}(\mathbb{P}, \mathbb{Q})$ has a dual representation

$$\mathcal{W}(\mathbb{P}, \mathbb{Q}) := \max_{D} \{\mathbb{E}_{\mathbf{x} \sim \mathbb{P}}[D(\mathbf{x})] - \mathbb{E}_{\mathbf{y} \sim \mathbb{Q}}[D(\mathbf{y})]\}, \quad (2)$$

where the maximization is over all 1-Lipschitz continuous functions.

**Sobolev space and Sobolev dual norm.** Let $L^2(\mathbb{P})$ be the Hilbert space on $\mathbb{R}^d$ equipped with an inner product $\langle u, v \rangle_{L^2(\mathbb{P})} := \int_{\mathbb{R}^d} uv d\mathbb{P}(\mathbf{x})$. The (weighted) Sobolev space $H^1$ is defined as the closure of $C_0^\infty$, a set of smooth functions on $\mathbb{R}^d$ with compact support, with respect to norm $\|u\|_{H^1} := \left(\int_{\mathbb{R}^d} (u^2 + \|\nabla u\|_2^2) d\mathbb{P}(\mathbf{x})\right)^{1/2}$, where $\mathbb{P}$ has a density. For $v \in L^2$, its Sobolev dual norm

$\|v\|_{H^{-1}}$ is defined by (Evans, 2010)

$$\|v\|_{H^{-1}} := \sup_{u \in H^1} \left\{ \langle v, u \rangle_{L^2} : \int_{\mathbb{R}^d} \|\nabla u\|_2^2 \, d\mathbb{P}(\mathbf{x}) \leq 1, \int_{\mathbb{R}^d} u(\mathbf{x}) d\mathbb{P}(\mathbf{x}) = 0 \right\}.$$

The constraint $\int_{\mathbb{R}^d} u(\mathbf{x}) d\mathbf{x} = 0$ is necessary to guarantee the finiteness of the supremum, and the supermum can be equivalently taken over $C_0^\infty$.

## 3 PROPOSED MODEL: STEIN BRIDGING

In this section, we formulate our model *Stein Bridging*. A scheme of our framework is illustrated in Figure 1. Denote by $\mathbb{P}_{\text{real}}$ the underlying real distribution from which the data $\{\mathbf{x}\}$ are sampled. The formulation simultaneously learns two generative models – one explicit and one implicit – that represent estimates of $\mathbb{P}_{\text{real}}$. The explicit generative model has a distribution $\mathbb{P}_E$ on $\mathcal{X}$ with explicit probability density proportional to $\exp(-E(\mathbf{x}))$, $\mathbf{x} \in \mathcal{X}$, where $E$ is referred to as an energy function. We focus on energy-based explicit model in model formulation as it

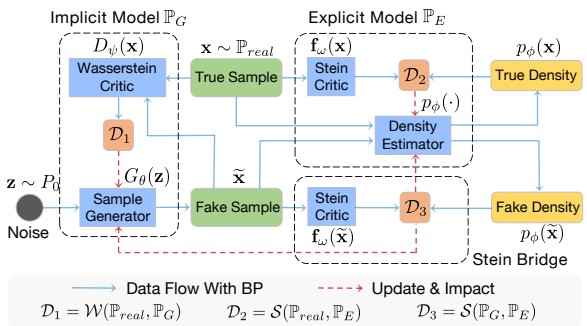

Figure 1: Model framework for *Stein Bridging*.

does not enforce any constraints or assume specific density forms. For specifications, one can also consider other explicit models, like autoregressive models or directly using some density forms such as Gaussian distribution with given domain knowledge. The implicit model transforms an easy-to-sample random noise $\mathbf{z}$ with distribution $P_0$ via a generator $G$ to a sample $\widetilde{x} = G(\mathbf{z})$ with distribution $\mathbb{P}_G$. Note that for distribution $\mathbb{P}_E$, we have its explicit density without normalizing term, while for $\mathbb{P}_G$ and $\mathbb{P}_{\text{real}}$, we have samples from two distributions. Hence, we can use the Stein discrepancy (that does *not* require the normalizing constant) as a measure of closeness between the explicit distribution $\mathbb{P}_E$ and the real distribution $\mathbb{P}_{\text{real}}$, and use the Wasserstein metric (that only requires only samples from two distributions) as a measure of closeness between the implicit distribution $\mathbb{P}_G$ and the real data distribution $\mathbb{P}_{\text{real}}$.

To jointly learn the two generative models $\mathbb{P}_G$ and $\mathbb{P}_E$, arguably the most straightforward way is to minimize the sum of the Stein discrepancy and the Wasserstein metric:

$$\min_{E,G} \mathcal{W}(\mathbb{P}_{\text{real}}, \mathbb{P}_G) + \lambda \mathcal{S}(\mathbb{P}_{\text{real}}, \mathbb{P}_E),$$

where $\lambda \geq 0$. However, this approach appears no different than learning the two generative models separately. To achieve information sharing between two models, we incorporate another term $\mathcal{S}(\mathbb{P}_G, \mathbb{P}_E)$ – called the *Stein bridge* – that measures the closeness between the explicit distribution $\mathbb{P}_E$ and the implicit distribution $\mathbb{P}_G$:

$$\min_{E,G} \mathcal{W}(\mathbb{P}_{\text{real}}, \mathbb{P}_G) + \lambda_1 \mathcal{S}(\mathbb{P}_{\text{real}}, \mathbb{P}_E) + \lambda_2 \mathcal{S}(\mathbb{P}_G, \mathbb{P}_E), \tag{3}$$

where $\lambda_1, \lambda_2 \geq 0$. The Stein bridge term in (3) pushes the two models to achieve a consensus.

**Remark 1**. Our formulation is flexible in choosing both the implicit and explicit models. In (3), we can choose statistical distances other than the Wasserstein metric $\mathcal{W}(\mathbb{P}_{\text{real}}, \mathbb{P}_G)$ to measure closeness between $\mathbb{P}_{\text{real}}$ and $\mathbb{P}_G$, such as Jensen-Shannon divergence, as long as its computation requires only samples from the involved two distributions. Hence, one can use GAN architectures other than WGAN to parametrize the implicit model. In addition, one can replace the first Stein discrepancy term $\mathcal{S}(\mathbb{P}_{\text{real}}, \mathbb{P}_E)$ in (3) by other statistical distances as long as its computation is efficient and hence other explicit models can be used. For instance, if the normalizing constant of $\mathbb{P}_E$ is known or easy to calculate, one can use Kullback-Leibler (KL) divergence.

**Remark 2**. The choice of the Stein discrepancy for the bridging term $\mathcal{S}(\mathbb{P}_G, \mathbb{P}_E)$ is crucial and cannot be replaced by other statistical distances such as KL divergence, since the data-generating distribution does not have an explicit density form (not even up to a normalizing constant). This

is exactly one important reason why Stein bridging was proposed, which requires only samples from the data distribution and only the log-density of the explicit model without the knowledge of normalizing constant as estimated in MCMC or other methods.

In our implementation, we parametrize the generator in implicit model and the density estimator in explicit model as $G_\theta(\mathbf{z})$ and $p_\phi(\mathbf{x})$, respectively. The Wasserstein term in (3) is implemented using its equivalent dual representation in (2) with a parametrized critic $D_\psi(\mathbf{x})$. The two Stein terms in (3) can be implemented using (1) with either a Stein critic (parametrized as a neural network, i.e., $\mathbf{f}_w(\mathbf{x})$), or the non-parametric Kernel Stein Discrepancy. Our implementation iteratively updates the explicit and implicit models. Details for model specifications and optimization are in Appendix E.2. We also compare with some related works that attempt to combine both of the worlds (such as energy-based GAN, contrastive learning and cooperative learning) in Appendix A.3.

## 4 THEORETICAL ANALYSIS

In this section, we theoretically show that the Stein bridge allows the two models to facilitate each other's training by imposing certain regularizations on both the implicit and the explicit models, as well as stabilizing the training dynamics.

### 4.1 REGULARIZATION VIA STEIN BRIDGE

We first show the regularization effect of the Stein bridge on the Wasserstein critic. Define the *kernel Sobolev dual norm* as

$$\|D\|_{H^{-1}(\mathbb{P};k)} := \sup_{u \in C_0^\infty} \{\langle D, u\rangle_{L^2(\mathbb{P})} : \mathbb{E}_{\mathbf{x}, \mathbf{x}' \sim \mathbb{P}}[\nabla u(\mathbf{x})^\top k(\mathbf{x}, \mathbf{x}')\nabla u(\mathbf{x}')] \leq 1, \ \mathbb{E}_{\mathbb{P}}[u] = 0\}.$$

which can be viewed as a kernel generalization of the Sobolev dual norm defined in Section 2, which reduces to the Sobolev dual norm when $k(\mathbf{x}, \mathbf{x}') = \mathbb{I}(\mathbf{x} = \mathbf{x}')$ and $\mathbb{P}$ is the Lebesgue measure.

**Theorem 1.** *Assume that $\{\mathbb{P}_G\}_G$ exhausts all continuous probability distributions and $\mathcal{S}$ is chosen as kernel Stein discrepancy. Then problem (3) is equivalent to*

$$\min_E \max_D \left\{ \mathbb{E}_{\mathbf{y} \sim \mathbb{P}_E}[D(\mathbf{y})] - \mathbb{E}_{\mathbf{x} \sim \mathbb{P}_{\text{real}}}[D(\mathbf{x})] - \frac{1}{4\lambda_2} \|D\|_{H^{-1}(\mathbb{P}_E;k)}^2 + \lambda_1 \mathcal{S}(\mathbb{P}_{\text{real}}, \mathbb{P}_E) \right\}.$$

The kernel Sobolev norm regularization penalizes large variation of the Wasserstein critic $D$. Particularly, observe that (Villani, 2008) if $k(\mathbf{x}, \mathbf{x}') = \mathbb{I}(\mathbf{x} = \mathbf{x}')$ and $\mathbb{E}_{\mathbb{P}_E}[D] = 0$, and then

$$\|D\|_{H^{-1}(\mathbb{P}_E;k)} = \lim_{\epsilon \to 0} \frac{\mathcal{W}_2((1 + \epsilon D)\mathbb{P}_E, \mathbb{P}_E)}{\epsilon},$$

where $\mathcal{W}_2$ denotes the 2-Wasserstein metric. Hence, the Sobolev dual norm regularization ensures $D$ not to change suddenly on high-density region of $\mathbb{P}_E$, and thus reinforces the learning of the Wasserstein critic. Stein bridge penalizes large variation of the Wasserstein critic, in the same spirit but of different form comparing to gradient-based penalty (e.g., (Gulrajani et al., 2017; Roth et al., 2017)). It prevents Wasserstein critic from being too strong during training and thus encourages mode exploration of sample generator. To illustrate this, we conduct a case study where we train a generator over the data sampled from a mixture of Gaussian ($\mu_1 = [-1, -1]$, $\mu_2 = [1, 1]$ and $\Sigma = 0.2\mathbf{I}$). In Fig. 2(a) we compare gradient norms of the Wasserstein critic when training the generator with and without the Stein bridge. As we can see, Stein bridge can help to reduce gradient norms through training, with a similar effect as WGAN-GP.

Moreover, the Stein bridge also plays a part in smoothing the output from Stein discrepancy and we show the result in the following theorem.

**Theorem 2.** *Assume $\{\mathbb{P}_G\}_G$ exhausts all continuous probability distributions, and the Stein class defining the Stein discrepancy is compact (in some linear topological space). Then problem (3) is equivalent to*

$$\min_E \left\{ \lambda_1 \mathcal{S}(\mathbb{P}_{\text{real}}, \mathbb{P}_E) + \lambda_2 \max_{\mathbf{f}} \mathbb{E}_{\mathbf{x} \sim \mathbb{P}_{\text{real}}}[(\mathcal{A}_{\mathbb{P}_E}\mathbf{f})_{\lambda_2}(\mathbf{x})] \right\},$$

*where $(\mathcal{A}_{\mathbb{P}_E}\mathbf{f})_{\lambda_2}(\cdot)$ denotes the (generalized) Moreau-Yosida regularization of the function $\mathcal{A}_{\mathbb{P}_E}\mathbf{f}$ with parameter $\lambda_2$, i.e., $(\mathcal{A}_{\mathbb{P}_E}\mathbf{f})_{\lambda_2}(\mathbf{x}) = \min_{\mathbf{y} \in \mathcal{X}}\{\mathcal{A}_{\mathbb{P}_E}\mathbf{f}(\mathbf{y}) + \frac{1}{\lambda_2}||\mathbf{x} - \mathbf{y}||\}.$*

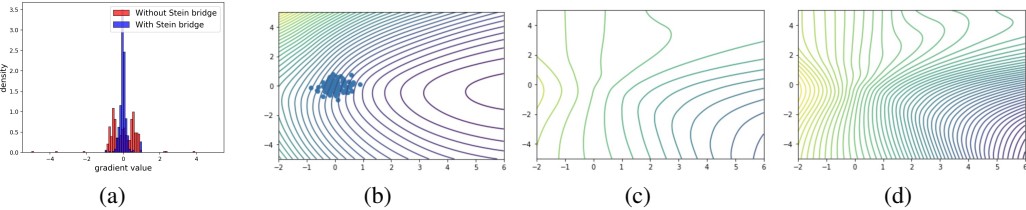

(a)        (b)        (c)        (d)

Figure 2: (a) The gradient norm of Wasserstein critic with (blue) and without (red) the Stein bridge when data are sampled from a mixture of Gaussian. (b) Contour of an energy model with one mode and empirical data from a distribution with a different mode (blue dots); (c) & (d) Contours of the Stein critics between the two distributions in (b) learned with and without the Stein bridge, respectively.

Note that $(\mathcal{A}_{\mathbb{P}_E}\mathbf{f})_{\lambda_2}$ is Lipschitz continuous with constant $1/\lambda_2$. Hence, the Stein bridge, together with the Wasserstein metric $\mathcal{W}(\mathbb{P}_{\mathrm{real}}, \mathbb{P}_G)$, plays as a Lipschitz regularization on the output of the Stein operator $\mathcal{A}_{\mathbb{P}_E}\mathbf{f}$ via Moreau-Yosida regularization. This suggests a novel regularization scheme for Stein-based GAN. By smoothing the Stein critic, the Stein bridge encourages the energy model to seek more modes in data instead of focusing on some dominated modes, thus alleviating mode-collapse issue. To illustrate this, we consider a case where we have an energy model initialized with one mode center and data sampled from distribution of another mode, as depicted in Fig. 2(b). Fig. 2(c) and 2(d) compare the Stein critics when using Stein bridge and not, respectively. The Stein bridge helps to smooth the Stein critic, as indicated by a less rapidly changing contour in Fig. 2(c) compared to Fig. 2(d), learned from the data and model distributions plotted in Fig. 2(b).

## 4.2 TRAINING STABILITY

In this subsection, we further show that Stein Bridging could help stabilize adversarial training between generator and Wasserstein critic with a local convergence guarantee. As is known, the training for minimax game in GAN is difficult. When using traditional gradient methods, the training would suffer from some oscillatory behaviors (Goodfellow, 2017; Liang & Stokes, 2019; Zhang & Yu, 2020). In order to better understand the optimization behaviors, we first compare the behaviors of WGAN, likelihood- and entropy-regularized WGAN, and our Stein Bridging under SGD via an easy to comprehend toy example in one-dimensional case. Fig. 3 shows numerical results that compare the optimization behaviors of above methods. As we can see, Stein Bridging achieves good convergence to the optimum point, while

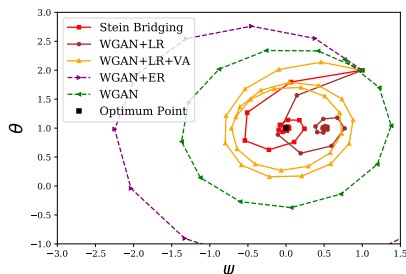

Figure 3: Numerical SGD updates of Stein Bridging, WGAN and its variants with different regularizations.

WGAN suffers from an oscillation instead of converging. Entropy regularization (ER) can encourage the generator to seek more modes but would make the model diverge in this case. By contrast, likelihood regularization (LR) can help for training stability but it changes the converging point to a biased distribution. A recently proposed variational annealing strategy (VA) (Tao et al., 2019) for regularized GAN introduces a trade-off between convergence and unbiased result. The detailed discussions and proofs are presented in Appendix D.1. We also generalize the convergence results to multi-dimensional bilinear system $F(\boldsymbol{\psi}, \boldsymbol{\theta}) = \boldsymbol{\theta}^\top \mathbf{A}\boldsymbol{\psi} - \mathbf{b}^\top \boldsymbol{\theta} - \mathbf{c}^\top \boldsymbol{\psi}$ in Appendix D.2. Our theoretical results indicate that Stein Bridging could stabilize the minimax training of GAN without changing its optimum. In the experiments, we will empirically validate our analysis.

## 5 EXPERIMENTS

In this section, we conduct experiments[1] to verify the effectiveness of proposed method from multifaceted views. We consider two synthetic datasets with mixtures of Gaussian distributions: Two-Circle and Two-Spiral. The first one is composed of 24 Gaussian mixtures that lie in two circles. Such dataset is extended from the 8-Gaussian-mixture scenario widely used in previous papers, so

---

[1]The experiment codes will be released.

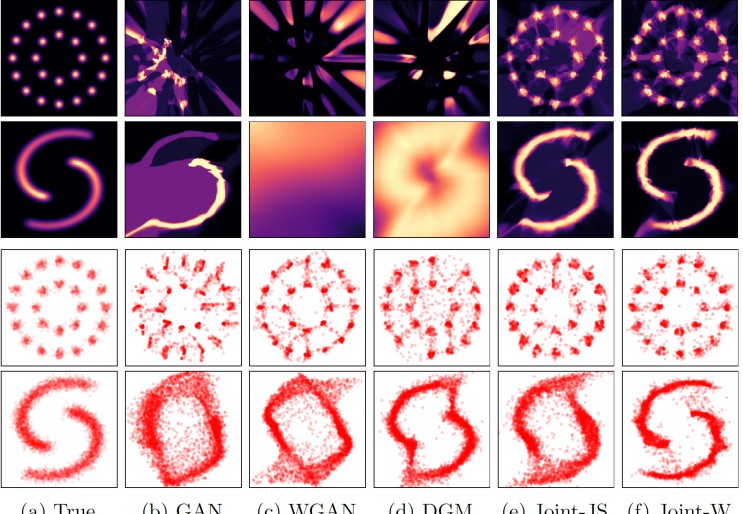

Figure 4: Comparison for density estimation. (a) True densities of real distribution and (b)∼(f) estimated densities given by the estimators of different methods on Two-Circle (upper line) and Two-Spiral (bottom line) datasets.

Figure 5: Comparison for generated sample quality. (a) True samples from real distribution and (b)∼(f) generated samples produced by the generators of different methods on Two-Circle (upper line) and Two-Spiral (bottom line) datasets.

(a) True      (b) GAN      (c) WGAN      (d) DGM      (e) Joint-JS      (f) Joint-W

that we can use it to test the quality of generated samples and mode coverage of learned energy. The second dataset consists of 100 Gaussian mixtures whose centers are densely arranged on two centrally symmetrical spiral-shaped curves. This dataset can be used to examine the power of generative model on complicated data distributions. The ground-truth distributions and samples are shown in Fig. 4(a) and Fig. 5(a). Furthermore, we also apply the method to MNIST and CIFAR datasets which require the model to deal with high-dimensional image data. In each dataset, we use observed samples as input of the model and leverage them to train our model. The details for each dataset are reported in Appendix E.1.

We term the model *Joint-W* if using Wasserstein metric in (3) and *Joint-JS* if using JS divergence in this section. We consider several competitors. For implicit generative models, we basically consider the counterparts without joint training with energy model, which are equivalently valina GAN and WGAN with gradient penalty (Gulrajani et al., 2017), for ablation study. Also, as comparison to the new regularization effects by Stein Bridging, we consider a recently proposed variational annealing regularization (Tao et al., 2019) for GANs (short as GAN+VA/WGAN+VA). We employ denoising auto-encoder to estimate the gradient for regularization penalty, which is proposed by (Alain & Bengio, 2014). For explicit models, we also consider the counterparts without joint training with generator model, i.e., directly training Deep Energy Model (DEM) using Stein discrepancy (Grathwohl et al., 2020). Besides we compare with energy calibrated GAN (EGAN) (Dai et al., 2017) and Deep Directed Generative (DGM) Model (Kim & Bengio, 2017) which adopt contrastive divergence to train a sample generator with an energy estimator. See Appendix A for brief introduction of these methods and Appendix E.3 for implementation details.

## 5.1 DENSITY ESTIMATION OF EXPLICIT MODEL

**Mode Coverage for Complicated Distributions.** One advantage of joint learning is that the generator could help the density estimator to capture more accurate distribution. As shown in Two-Circle case in Fig 5, both Joint-JS and Joint-W manage to capture all Gaussian components while other methods miss some of modes. In Two-Spiral case in Fig 4, Joint-JS and Joint-W exactly fit the ground-truth distribution. Nevertheless, DEM misses one spiral while EGAN degrades to a uniform-like distribution. DGM manages to fit two spirals but allocate high densities to regions that have low densities in the groung-truth distribution. As quantitative comparison, we study three evaluation metrics: KL & JS divergence and Area Under the Curve (AUC). The detailed information and results are given in Appendix E.4 and Table 5 respectively. The values show that Joint-W and Joint-JS provide better density estimation than all competitors over a large margin .

**Density Rankings for High-Dimensional Digits.** We also rank generated digits (and true digits) on MNIST w.r.t densities given by the energy model in Fig. 11, Fig. 12 and Fig. 13. As depicted in the figures, the digits with high densities (or low densities) given by Joint-JS possess enough diversity (the thickness, the inclination angles as well as the shapes of digits diverses). By constrast, all the digits with high densities given by DGM tend to be thin and digits with low densities are very thick.

Also, as for EGAN, digits with high (or low) densities appear to have the same inclination angle (for high densities, '1' keeps straight and '9' 'leans' to the left while for low densities, just the opposite), which indicates that DGM and EGAN tend to allocate high (or low) densities to data with certain modes and miss some modes that possess high densities in ground-truth distributions. By contrast, our method manages to capture these complicated features in data distributions.

**Detection for Out-of-distribution Samples.** We further study model performance on detection for out-of-distribution samples. We consider CIFAR-10 images as positive samples and construct negative samples by (I) flip images, (II) add random noise, (III) overlay two images and (IV) use images from LSUN dataset, respectively. A good density models trained on CIFAR-10 are expected to give high densities to positive samples and low densities to negative samples, with exception for case (I) (flipping images are not exactly negative samples and the model should give high densities). We use the density values rank samples and calculate AUC of false positive rate v.s. true positive rate, reported in Table 2. Our model Joint-W manages to distinguish samples for (II), (III), (IV) and is not fooled by flipping images, while DEM and EGAN fail to detect out-of-distribution samples and DGM recognizes flipping images as negative samples.

## 5.2 SAMPLE QUALITY OF IMPLICIT MODEL

**Generated Samples over Synthetic Datasets.** Calibrating explicit (unnormalized) density model with implicit generator is expected to improve the quality of generated samples. In Fig. 5 we show the results of different generators in Two-Circle and Two-Spiral datasets. In Two-Circle, there are a large number of generated samples given by GAN, WGAN-GP and DGM locating between two Gaussian components, and the boundary for each component is not distinguishable. Since the ground-truth

Table 1: Inception Scores (IS) and Fréchet Inception Distance (FID) on CIFAR-10.

| Method | IS | FID |
|---|---|---|
| WGAN-GP | 6.74±0.041 | 42.2±0.572 |
| Energy GAN | 6.89±0.081 | 45.6±0.375 |
| WGAN+VA | 6.90±0.058 | 45.3±0.307 |
| DGM | 6.51±0.041 | 48.8±0.492 |
| **Joint-W(ours)** | **7.12**±0.101 | **41.0**±0.546 |

densities of regions between two components are very low, such generated samples possess low-quality, which depicts that these models capture the combinations of two dominated features (i.e., modes) in data but such combination makes no sense in practice. By contrast, Joint-JS and Joint-W could alleviate such issue, reduce the low-quality samples and produce more distinguishable boundaries. In Two-Spiral, similarly, the generated samples given by GAN and WGAN-GP form a circle instead of two spirals while the samples of DGM 'link' two spirals. Joint-JS manages to focus more on true high densities compared to GAN and Joint-W provides the best results. To quantitatively measure the sample quality, we adopt Maximum Mean Discrepancy (MMD) and High-quality Sample Rate (HSR). The details are in Appendix E.4 and we report results in Table 5 where our models significantly outperform the competitors over a large margin.

**Sample Quality for Generated Images.** We calculate the Inception Score (IS) and Fréchet Inception Distance (FID) to measure the sample quality on CIFAR-10. As shown in Table 1, Joint-W outperforms other competitors by 0.2 and achieves $5.6\%$ improvement over WGAN-GP w.r.t IS. As for FID, Joint-W slightly outperforms WGAN-GP and beats energy-based GAN and variational annealing regularized WGAN over a large margin. One possible reason is that these methods both consider entropy regularization which encourages diversity of generated samples but will have a negative effect on sample quality. Stein Bridging can overcome this issue via joint training with explicit model. The performance of DGM tends to be much worse than others. In practice, DGM is hard for convergence and suffers from severe instability in training.

**Model Performance in Contaminated or Limited Data.** As further discussions, we highlight that Stein Bridging has promising power in some extreme cases where the training sample are contaminated or limited. We consider noised data scenario and randomly add $n$ noise points sampled from Gaussian distribution $\mathcal{N}(\mathbf{0}, \sigma_0 \mathbf{I})$ where $\sigma_0 = 2$ to the original true samples in Two-Circle dataset. The results on noised dataset are presented in Fig. 8(a) where we set noise ratio $n = [40, 100, 160, 300, 400, 600, 800, 1000]$ and report the HSRs of Joint-W and WGAN-GP. The noise ratio in data impacts the performance of WGAN-GP and Joint-W, but comparatively, the performance decline of Joint-W is less insignificant than WGAN-GP, which indicates better robustness of joint training w.r.t. noised data.

Table 2: AUCs for out-of-distribution sample detection on CIFAR-10. We use negative samples from (I) flip images, (II) add random noise, (III) overlay two images and (IV) use images from LSUN dataset.

| AUC | I | II | III | IV |
|---|---|---|---|---|
| Joint-W | 0.50 | 0.92 | 0.95 | 0.85 |
| DEM | 0.50 | 0.52 | 0.51 | 0.56 |
| DGM | 1.00 | 1.00 | 1.00 | 0.82 |
| EGAN | 0.50 | 0.42 | 0.30 | 0.52 |

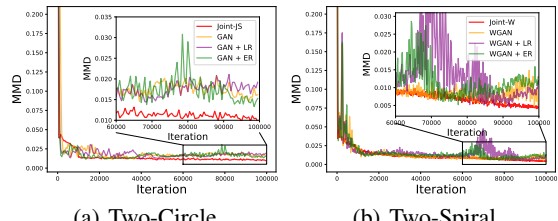

(a) Two-Circle      (b) Two-Spiral

Figure 6: Learning curves of Joint-W (resp. Joint-JS) compared with WGAN (resp. GAN) and its regularization-based variants on Two-Circle and Two-Spiral datasets.

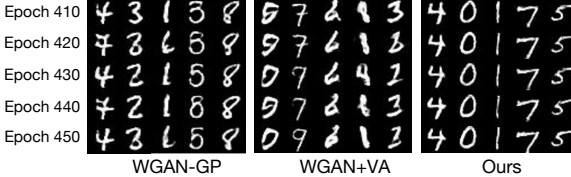

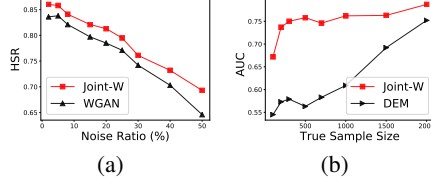

(a)      (b)

Figure 7: Generated digits given by the same noise $z$ in adjacent training epochs on MNIST.

Figure 8: Impact of (a) noise in data and (b) insufficient data on model performance.

To study the impact of insufficient data, in Fig. 8(b), we consider sample size $N_2$ as $[100, 200, 300, 500, 700, 1000, 2000]$ in Two-Spiral dataset and report the AUC of Joint-W and DEM. When sample size decreases from 2000 to 100, the AUC value of DEM declines dramatically, showing its dependency on sufficient training samples. By contrast, the AUC of Joint-W exhibits a small decline when the sample size is more than 500 and suffers from an obvious decline when it is less than 300. Such phenomenon demonstrates its lower sensitivity to data size.

## 5.3 ENHANCING THE STABILITY OF GAN

Joint training also helps to stabilize training dynamics. In Fig. 6 we present the learning curves of Joint-W (resp. Joint-JS) compared with WGAN (resp. GAN) and likelihood- and entropy-regularized WGAN (resp. GAN). The curves depict that joint training could reduce the variance of metric values especially during the second half of training. Furthermore, we visualize generated digits given by the same noise $z$ in adjacent epochs in Fig. 7. The results show that Joint-W gives more stable generation in adjacent epochs while generated samples given by WGAN-GP and WGAN+VA exhibit an obvious variation. Especially, some digits generated by WGAN-GP and WGAN+VA change from one class to another, which is quite similar to the oscillation without convergence discussed in Section 3.2. To quantify the evaluation of bias in model distributions, we calculate distances between the means of 50000 generated digits (resp. images) and 50000 true digits (resp. images) in MNIST (reps. CIFAR-10). The results are reported in Table 4. We can see that the model distributions of other competitors are more biased from true data distribution, compared with Joint-W.

## 6 CONCLUSIONS

In this paper, we aim at uniting the training for implicit generative model (represented by GAN or WGAN) and explicit generative model (represented by a deep energy-based model) via an bridging term of Stein discrepancy between the generator and the energy-based density estimator. Theoretically, we show that joint training could i) enforce dual regularization effects on both models and thus encourage mode exploration, and ii) help to facilitate the convergence of minimax training dynamics. We also conduct extensive experiments on different tasks and applications to verify our theoretical findings as well as demonstrate the superiority of our method compared with training generator models or energy-based models alone. Our formulation is flexible in handling various implicit or explicit models. As such, for future works, one can try other generative models such as VAE or flowed-based model as replacement for our GAN and energy-based models. It would also be interesting to exploit our formulation in the context of few-shot learning in generative models.

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

# A LITERATURE REVIEWS

We discuss some of related literature and shed lights on the relationship between our work with others.

## A.1 EXPLICIT GENERATIVE MODELS

Explicit generative models are interested in fitting each instance with a scalar (unnormalized) density expected to explicitly capture the distribution behind data. Such densities are often up to a constant and called as energy functions which are common in undirected graphical models (LeCun et al., 2006). Hence, explicit generative models are also termed as energy-based models. An early version of energy-based models is the FRAME (Filters, Random field, And Maximum Entropy) model (Zhu et al., 1997; Wu et al., 2000). Later on, some works leverage deep neural networks to model the energy function (Ngiam et al., 2011; Xie et al., 2016b) and pave the way for researches on deep energy model (DEM) (e.g., (Liu & Wang, 2017; Kim & Bengio, 2017; Zhai et al., 2016; Haarnoja et al., 2017; Du & Mordatch, 2019; Nijkamp et al., 2019)). Apart from DEM, there are also some other forms of deep explicit models based on restricted Boltzmann machines like deep belief networks (Hinton et al., 2006) and deep Boltzmann machines (Salakhutdinov & Hinton, 2009).

The normalized constant under the energy function requires an intractable integral over all possible instances, which makes the model hard to learn via Maximum Likelihood Estimation (MLE). To solve this issue, some works propose to approximate the constant by MCMC methods (Geman & Geman, 1984; Neal, 2011). However, MCMC requires an inner-loop samples in each training, which induces high computational costs. Another solution is to optimize an alternate surrogate loss function. For example, contrastive divergence (CD) (Liu & Wang, 2017) is proposed to measure how much KL divergence can be improved by running a small numbers of Markov chain steps towards the intractable likelihood, while score matching (SM) (Hyvärinen, 2005) detours the constant by minimizing the distance for gradients of log-likelihoods. A recent study (Grathwohl et al., 2020) uses Stein discrepancy to train unnormalized model. The Stein discrepancy does not require the normalizing constant and makes the training tractable. Moreover, the intractable normalized constant makes it hard to sample from. To obtain an accurate samples from unnormalized densities, many studies propose to approximate the generation by diffusion-based processes, like generative flow (Nguyen et al., 2017) and variational gradient descent ((Liu & Wang, 2016)). Also, a recent work (Hu et al., 2018) leverages Stein discrepancy to design a neural sampler from unnormalized densities. The fundamental disadvantage of explicit model is that the energy-based learning is difficult to accurately capture the distribution of true samples due to the low manifold of real-world instances (Liu & Wang, 2017).

## A.2 IMPLICIT GENERATIVE MODELS

Implicit generative models focus on a generation mapping from random noises to generated samples. Such mapping function is often called as generator and possesses better flexibility compared with explicit models. Two typical implicit models are Variational Auto-Encoder (VAE) (Kingma & Welling, 2014) and Generative Adversarial Networks (GAN) (Goodfellow et al., 2014). VAE introduces a latent variable and attempts to maximize the variational lower bound for likelihood of joint distribution of latent variable and observable variable, while GAN targets an adversarial game between the generator and a discriminator (or critic in WGAN) that aims at discriminating the generated and true samples. In this paper, we focus on GAN and its variants (e.g., WGAN (Arjovsky et al., 2017), WGAN-GP (Gulrajani et al., 2017), DCGAN (Radford et al., 2016), etc.) as the implicit generative model and we leave the discussions on VAE as future work.

Two important issues concerning GAN and its variants are instability of training and local optima. The typical local optima for GAN can be divided into two categories: mode-collapse (the model fails to capture all the modes in data) and mode-redundance (the model generates modes that do not exist in data). Recently there are many attempts to solve these issues from various perspectives. One perspective is from regularization. Two typical regularization methods are likelihood-based and entropy-based regularization with the prominent examples (Warde-Farley & Bengio, 2017) and (Li & Turner, 2018) that respectively leverage denoising feature matching and implicit gradient approximation to enforce the regularization constraints. The likelihood and entropy regularizations could

respectively help the generator to focus on data distribution and encourage more diverse samples, and a recent work (Tao et al., 2019) uses Langevin dynamics to indicate that i) the entropy and likelihood regularizations are equivalent and share an opposite relationship in mathematics, and ii) both regularizations would make the model converge to a surrogate point with a bias from original data distribution. Then (Tao et al., 2019) proposes a variational annealing strategy to empirically unite two regularizations and tackle the biased distributions.

To deal with the instability issue, there are also some recent literatures from optimization perspectives and proposes different algorithms to address the non-convergence of minimax game optimization (for instance, (Gemp & Mahadevan, 2018; Liang & Stokes, 2019; Gidel et al., 2019)). Moreover, the disadvantage of implicit models is the lack of explicit densities over instances, which disables the black-box generator to characterize the distributions behind data.

### A.3 Attempts to Combine Both of the Worlds

Recently, there are several studies that attempt to combine explicit and implicit generative models from different ways. For instance, (Zhao et al., 2017) proposes energy-based GAN that leverages energy model as discriminator to distinguish the generated and true samples. The similar idea is also used by (Kim & Bengio, 2017) and (Dai et al., 2017) which let the discriminator estimate a scaler energy value for each sample. Such discriminator is optimized to give high energy to generated samples and low energy to true samples while the generator aims at generating samples with low energy. The fundamental difference is that (Zhao et al., 2017) and (Dai et al., 2017) both aim at minimizing the discrepancy between distributions of generated and true samples while the motivation of (Kim & Bengio, 2017) is to minimize the KL divergence between estimated densities and true samples. (Kim & Bengio, 2017) adopts contrastive divergence (CD) to link MLE for energy model over true data with the adversarial training of energy-based GAN. However, both CD-based method and energy-based GAN have limited power for both generator and discriminator. Firstly, if the generated samples resemble true samples, then the gradients for discriminator given by true and generated samples are just the opposite and will counteract each other, and the training will stop before the discriminitor captures accurate data distribution. Second, since the objective boils down to minimizing the KL divergence (for (Kim & Bengio, 2017)) or Wasserstein distance (for (Dai et al., 2017)) between model and true distributions, the issues concerning GAN (or WGAN) like training instability and mode-collapse would also bother these methods.

Another way for combination is by cooperative training. (Xie et al., 2016a) (and its improved version (Xie et al., 2018)) leverages the samples of generator as the MCMC initialization for energy-based model. The synthesized samples produced from finite-step MCMC are closer to the energy model and the generator is optimized to make the finite-step MCMC revise its initial samples. Also, a recent work (Du et al., 2018) proposes to regard the explicit model as a teacher net who guides the training of implicit generator as a student net to produce samples that could overcome the mode-collapse issue. The main drawback of cooperative training is that they indirectly optimize the discrepancy between the generator and data distribution via the energy model as a 'mediator', which leads to a fact that once the energy model gets stuck in a local optimum (e.g., mode-collapse or mode-redundance) the training for the generator would be affected. In other words, the training for two models would constrain rather than exactly compensate each other. Different from existing methods, our model considers three discrepancies simultaneously as a triangle to jointly train the generator and the estimator, enabling them to compensate and reinforce each other.

## B Background for Stein Discrepancy

Assume $q(\mathbf{x})$ to be a continuously differentiable density supported on $\mathcal{X} \subset \mathbb{R}^d$ and $\mathbf{f} : \mathbb{R}^d \to \mathbb{R}^{d'}$ a smooth vector function. Define $\mathcal{A}_q[\mathbf{f}(\mathbf{x})] = \nabla_{\mathbf{x}} \log q(\mathbf{x}) \mathbf{f}(\mathbf{x})^\top + \nabla_{\mathbf{x}} \mathbf{f}(\mathbf{x})$ as a Stein operator. If $\mathbf{f}$ is a Stein class (satisfying some mild boundary conditions) then we have the following Stein identity property:

$$\mathbb{E}_{\mathbf{x} \sim q}[A_q[\mathbf{f}(\mathbf{x})]] = \mathbb{E}_{\mathbf{x} \sim q}[\nabla_{\mathbf{x}} \log q(\mathbf{x}) \mathbf{f}(\mathbf{x})^\top + \nabla_{\mathbf{x}} \mathbf{f}(\mathbf{x})] = 0.$$

Such property induces Stein discrepancy between distributions $\mathbb{P} : p(\mathbf{x})$ and $\mathbb{Q} : q(\mathbf{x})$, $\mathbf{x} \in \mathcal{X}$:

$$\mathcal{S}(\mathbb{Q}, \mathbb{P}) = \sup_{\mathbf{f} \in \mathcal{F}} \{\mathbb{E}_{\mathbf{x} \sim q}[A_p[\mathbf{f}(\mathbf{x})]] = \sup_{\mathbf{f} \in \mathcal{F}} \{\Gamma(\mathbb{E}_{\mathbf{x} \sim q}[\nabla_{\mathbf{x}} \log p(\mathbf{x}) \mathbf{f}(\mathbf{x})^\top + \nabla_{\mathbf{x}} \mathbf{f}(\mathbf{x})])\}, \quad (4)$$

where $\mathbf{f}$ is what we call *Stein critic* that exploits over function space $\mathcal{F}$ and if $\mathcal{F}$ is large enough then $\mathcal{S}(\mathbb{Q}, \mathbb{P}) = 0$ if and only if $\mathbb{Q} = \mathbb{P}$. Note that in (1), we do not need the normalized constant for $p(\mathbf{x})$ which enables Stein discrepancy to deal with unnormalized density.

If $\mathcal{F}$ is a unit ball in a Reproducing Kernel Hilbert Space (RKHS) with a positive definite kernel function $k(\cdot, \cdot)$, then the supremum in (1) would have a close form (see (Liu et al., 2016; Chwialkowski et al., 2016; Oates et al., 2017) for more details):

$$\mathcal{S}_K(\mathbb{Q}, \mathbb{P}) = \mathbb{E}_{\mathbf{x}, \mathbf{x}' \sim q}[u_p(\mathbf{x}, \mathbf{x}')], \tag{5}$$

where $u_p(\mathbf{x}, \mathbf{x}') = \nabla_{\mathbf{x}} \log p(\mathbf{x})^\top k(\mathbf{x}, \mathbf{x}') \nabla_{\mathbf{x}} \log p(\mathbf{x}') + \nabla_{\mathbf{x}} \log p(\mathbf{x})^\top \nabla_{\mathbf{x}} k(x, \mathbf{x}') + \nabla_{\mathbf{x}} k(\mathbf{x}, \mathbf{x}')^\top \nabla_{\mathbf{x}} \log p(\mathbf{x}') + tr(\nabla_{\mathbf{x}, \mathbf{x}'} k(\mathbf{x}, \mathbf{x}'))$. The (5) gives the Kernel Stein Discrepancy (KSD).

## C  PROOFS OF RESULTS IN SECTION 4.1

### C.1  PROOF OF THEOREM 1

*Proof.* Applying Kantorovich's duality on $\mathcal{W}(\mathbb{P}_G, \mathbb{P}_r)$ and using the exhaustiveness assumption on the generator, we rewrite the problem as

$$\min_{E, \mathbb{P}} \max_D \{\mathbb{E}_{\mathbb{P}}[D] - \mathbb{E}_{\mathbb{P}_{\text{real}}}[D] + \lambda_1 \mathcal{S}(\mathbb{P}_{\text{real}}, \mathbb{P}_E) + \lambda_2 \mathcal{S}(\mathbb{P}, \mathbb{P}_E)\}, \tag{6}$$

where the minimization with respect to $E$ is over all energy functions, the minimization with respect to $\mathbb{P}$ is over all probability distributions with continuous density, and the maximization with respect to $D$ is over all 1-Lipschitz continuous functions. Recall the definition of kernel Stein discrepancy

$$\mathcal{S}(\mathbb{P}, \mathbb{P}_E) = \mathbb{E}_{\mathbf{x}, \mathbf{x}' \sim \mathbb{P}}[(\nabla_x \log d\mathbb{P}/d\mathbb{P}_E(\mathbf{x}))^\top k(\mathbf{x}, \mathbf{x}') \nabla_x \log d\mathbb{P}/d\mathbb{P}_E(\mathbf{x}')],$$

where $d\mathbb{P}/d\mathbb{P}_E$ is the Radon-Nikodym derivative. Observe that $\mathcal{S}(\mathbb{P}, \mathbb{P}_E)$ is infinite if $\mathbb{P}$ is not absolutely continuous with respect to $\mathbb{P}_E$. Hence, to minimize the objective of (6), it suffices to consider those $\mathbb{P}$'s that are absolutely continuous with respect to $\mathbb{P}_E$. Introducing a variable replacement $h(\mathbf{x}) := d\mathbb{P}/d\mathbb{P}_E(\mathbf{x}) - 1$, then problem (6) becomes

$$\min_{E, h} \max_D \left\{ \mathbb{E}_{\mathbb{P}_E}[(1 + h)D] - \mathbb{E}_{\mathbb{P}_{\text{real}}}[D] + \lambda_1 \mathcal{S}(\mathbb{P}_{\text{real}}, \mathbb{P}_E) \right.$$
$$\left. + \lambda_2 \cdot \mathbb{E}_{\mathbf{x}, \mathbf{x}' \sim \mathbb{P}}[\nabla_x \log(1 + h(\mathbf{x}))^\top k(\mathbf{x}, \mathbf{x}') \nabla_x \log(1 + h(\mathbf{x}'))] \right\}, \tag{7}$$

where the minimization with respect to $h$ is over all $L^1(\mathbb{P}_E)$ functions with $\mathbb{P}_E$-expectation zero.

Fixing $E$, we claim that we can swap $\min_h$ and $\max_D$. Indeed, without loss of generality, we can restrict $D$ to be such that $D(\mathbf{x}_0) = 0$ for some element $\mathbf{x}_0$, as a constant shift does not change the value of $\mathbb{E}_{\mathbb{P}_E}[(1 + h)D] - \mathbb{E}_{\mathbb{P}_{\text{real}}}[D]$. The set of Lipschitz functions that vanish at $\mathbf{x}_0$ is a Banach space, and the set of 1-Lipschitz functions is compact (Weaver, 1999). Moreover, $L^1(\mathbb{P}_E)$ is also a Banach space and the objective function is linear in both $h$ and $D$. The above verifies the condition of Sion's minimax theorem, and thus the claim is proved.

Swapping $\min_h$ and $\max_D$ in (7) and fixing $E$ and $D$, we consider

$$\min_{h: \mathbb{E}_{\mathbb{P}_E}[h] = 0} \{\mathbb{E}_{\mathbb{P}_E}[hD] + \lambda_2 \cdot \mathbb{E}_{\mathbf{x}, \mathbf{x}' \sim \mathbb{P}}[\nabla_x \log(1 + h(\mathbf{x}))^\top k(\mathbf{x}, \mathbf{x}') \nabla_x \log(1 + h(\mathbf{x}'))]\}$$

$$= \min_{h: \mathbb{E}_{\mathbb{P}_E}[h] = 0} \left\{ \mathbb{E}_{\mathbb{P}_E}[hD] + \lambda_2 \cdot \mathbb{E}_{\mathbf{x}, \mathbf{x}' \sim \mathbb{P}} \left[ \frac{\nabla_x h(\mathbf{x})^\top}{1 + h(\mathbf{x})} k(\mathbf{x}, \mathbf{x}') \frac{\nabla_x h(\mathbf{x}')}{1 + h(\mathbf{x}')} \right] \right\}$$

$$= \min_{h: \mathbb{E}_{\mathbb{P}_E}[h] = 0} \left\{ \mathbb{E}_{\mathbb{P}_E}[hD] + \lambda_2 \cdot \mathbb{E}_{\mathbf{x}, \mathbf{x}' \sim \mathbb{P}_E} \left[ \nabla_x h(\mathbf{x})^\top k(\mathbf{x}, \mathbf{x}') \nabla_x h(\mathbf{x}') \right] \right\},$$

where the first equality follows from the chain rule of the derivative, and the second equality follows from a change of measure $d\mathbb{P} = (1 + h)d\mathbb{P}_E$. Introducing an auxiliary variable $r$ so that $r^2$ is an

upper bound of $\mathbb{E}_{\mathbf{x},\mathbf{x}'\sim\mathbb{P}_E}\left[\nabla_x h(\mathbf{x})^\top k(\mathbf{x},\mathbf{x}')\nabla_x h(\mathbf{x}')\right]$, we have that

$$
\min_{h:\mathbb{E}_{\mathbb{P}_E}[h]=0}\left\{\mathbb{E}_{\mathbb{P}_E}[hD]+\lambda_2\cdot\mathbb{E}_{\mathbf{x},\mathbf{x}'\sim\mathbb{P}_E}\left[\nabla_x h(\mathbf{x})^\top k(\mathbf{x},\mathbf{x}')\nabla_x h(\mathbf{x}')\right]\right\}
$$

$$
=\min_{r\geq 0}\min_{h:\mathbb{E}_{\mathbb{P}_E}[h]=0}\left\{\mathbb{E}_{\mathbb{P}_E}[hD]+\lambda_2 r^2:\mathbb{E}_{\mathbf{x},\mathbf{x}'\sim\mathbb{P}_E}\left[\nabla_x h(\mathbf{x})^\top k(\mathbf{x},\mathbf{x}')\nabla_x h(\mathbf{x}')\right]\leq r^2\right\}
$$

$$
=\min_{r\geq 0}\min_{h:\mathbb{E}_{\mathbb{P}_E}[h]=0}\left\{r\mathbb{E}_{\mathbb{P}_E}[hD]+\lambda_2 r^2:\mathbb{E}_{\mathbf{x},\mathbf{x}'\sim\mathbb{P}_E}\left[\nabla_x h(\mathbf{x})^\top k(\mathbf{x},\mathbf{x}')\nabla_x h(\mathbf{x}')\right]\leq 1\right\}
$$

$$
=\min_{r\geq 0}\left\{\lambda_2 r^2-r\left\|D\right\|_{H^{-1}(\mathbb{P}_E;k)}\right\}
$$

$$
=-\frac{1}{4\lambda_2}\left\|D\right\|^2_{H^{-1}(\mathbb{P}_E;k)},
$$

where the first equality holds because $\mathbb{E}_{\mathbf{x},\mathbf{x}'\sim\mathbb{P}_E}\left[\nabla_x (rh)(\mathbf{x})^\top k(\mathbf{x},\mathbf{x}')\nabla_x (rh)(\mathbf{x}')\right]=r^2\mathbb{E}_{\mathbf{x},\mathbf{x}'\sim\mathbb{P}_E}\left[\nabla_x h(\mathbf{x})^\top k(\mathbf{x},\mathbf{x}')\nabla_x h(\mathbf{x}')\right]$ for all $r\geq 0$ and by introducing an auxiliary variable $r^2=\mathbb{E}_{\mathbf{x},\mathbf{x}'\sim\mathbb{P}_E}\left[\nabla_x h(\mathbf{x})^\top k(\mathbf{x},\mathbf{x}')\nabla_x h(\mathbf{x}')\right]$; the second equality follows from a change of variable from $h$ to $rh$; and the third equality follows from the definition of the kernel Sobolev dual norm. Plugging back in (7) yields the ideal result. $\qquad\square$

## C.2 PROOF FOR THEOREM 2

*Proof.* Applying the definition of Stein discrepancy on $\mathcal{S}(\mathbb{P}_E,\mathbb{P}_G)$ and under the exhaustiveness assumption of $G$, we rewrite the problem as

$$
\min_{E,\mathbb{P}}\max_{\mathbf{f}}\{\lambda_1\mathcal{S}(\mathbb{P}_{\text{real}},\mathbb{P}_E)+\lambda_2\mathbb{E}_{\mathbf{y}\sim\mathbb{P}}[\mathcal{A}_{\mathbb{P}_E}\mathbf{f}(\mathbf{y})]+\mathcal{W}(\mathbb{P}_{\text{real}},\mathbb{P})\},
$$

where the minimization with respect to $E$ is over the set of all engergy functions; the minimization with respect to $\mathbb{P}$ is over all continuous distributions; and the maximization with respect to $\mathbf{f}$ is over the Stein class for $\mathbb{P}_E$. Let us fix $E$. Using a similar argument as in the proof of Theorem 1, it suffices to restrict $\mathbb{P}$ on the set of distributions that are absolutely continuous with respect to $\mathbb{P}_E$, which can be identified as the set of $L^1(\mathbb{P}_E)$ functions with $\mathbb{P}_E$-mean zero and is thus Banach. Together with the compactness assumption of the Stein class, using Sion's minimax theorem, we can swap the minimization over $\mathbb{P}$ and the maximization over $\mathbf{f}$. Now, fixing further $\mathbf{f}$, consider

$$
\min_{\mathbb{P}}\left\{\lambda_2\mathbb{E}_{\mathbf{y}\sim\mathbb{P}}[\mathcal{A}_{\mathbb{P}_E}\mathbf{f}(\mathbf{y})]+\mathcal{W}(\mathbb{P}_{\text{real}},\mathbb{P})\right\}. \tag{8}
$$

Recall the definition of Wasserstein metric

$$
\mathcal{W}(\mathbb{P}_{\text{real}},\mathbb{P})=\min_{\gamma}\mathbb{E}_{(\mathbf{x},\mathbf{y})\sim\gamma}[\|\mathbf{x}-\mathbf{y}\|],
$$

where the minimization is over all joint distributions of $(\mathbf{x},\mathbf{y})$ with $\mathbf{x}$-marginal $\mathbb{P}_{\text{real}}$ and $\mathbf{y}$-marginal $\mathbb{P}$. We rewrite problem (8) as

$$
\min_{\mathbb{P},\gamma}\{\mathbb{E}_{(\mathbf{x},\mathbf{y})\sim\gamma}\left[\lambda_2\,\mathcal{A}_{\mathbb{P}_E}\mathbf{f}(\mathbf{y})+\|\mathbf{x}-\mathbf{y}\|\right]\},
$$

where $\gamma$ has marginals $\mathbb{P}_{\text{real}}$ and $\mathbb{P}$. Since $\mathbb{P}$ is unconstrained, the above problem is further equivalent to

$$
\min_{\gamma}\{\mathbb{E}_{(\mathbf{x},\mathbf{y})\sim\gamma}\left[\lambda_2\mathcal{A}_{\mathbb{P}_E}\mathbf{f}(\mathbf{y})+\|\mathbf{x}-\mathbf{y}\|\right]\},
$$

where the minimization is over all joint distributions of $(\mathbf{x},\mathbf{y})$ with $\mathbf{x}$-marginal being $\mathbb{P}_{\text{real}}$. Using the law of total expectation, the problem above is equivalent to

$$
\min_{\{\gamma_{\mathbf{x}}\}_{\mathbf{x}\in\text{supp}\mathbb{P}_{\text{real}}}}\mathbb{E}_{\mathbf{x}\sim\mathbb{P}_{\text{real}}}\left[\mathbb{E}_{\mathbf{y}\sim\gamma_{\mathbf{x}}}\left[\lambda_2\mathcal{A}_{\mathbb{P}_E}\mathbf{f}(\mathbf{y})+\|\mathbf{x}-\mathbf{y}\|\mid\mathbf{x}\right]\right]
$$

$$
=\mathbb{E}_{\mathbf{x}\sim\mathbb{P}_{\text{real}}}\left[\min_{\gamma_{\mathbf{x}}}\left\{\mathbb{E}_{\mathbf{y}\sim\gamma_{\mathbf{x}}}\left[\lambda_2\mathcal{A}_{\mathbb{P}_E}\mathbf{f}(\mathbf{y})+\|\mathbf{x}-\mathbf{y}\|\mid\mathbf{x}\right]\right\}\right]
$$

$$
=\mathbb{E}_{\mathbf{x}\sim\mathbb{P}_{\text{real}}}\left[\min_{\mathbf{y}\in\mathcal{X}}\{\lambda_2\mathcal{A}_{\mathbb{P}_E}\mathbf{f}(\mathbf{y})+\|\mathbf{x}-\mathbf{y}\|\}\right]
$$

where the minimization in the first line of the equation is over $\gamma_{\mathbf{x}}$, the set of all conditional distributions of $\mathbf{y}$ given $\mathbf{x}$ where $\mathbf{x}$ is over the support $\text{supp }\mathbb{P}_{\text{real}}$ of $\mathbb{P}_{\text{real}}$; the exchanging of $\min$ and $\mathbb{E}$

in the first equality follows from the interchangebability principle (Shapiro et al., 2009); the second equality holds because the infimum can be restricted to the set of point masses. Finally, the original problem is equivalent to

$$\min_E \max_{\mathbf{f}} \left\{ \lambda_1 \mathcal{S}(\mathbb{P}_{\text{real}}, \mathbb{P}_E) + \mathbb{E}_{\mathbf{x} \sim \mathbb{P}_{\text{real}}} \left[ \min_{\mathbf{y} \in \mathcal{X}} \{ \lambda_2 \mathcal{A}_{\mathbb{P}_E} \mathbf{f}(\mathbf{y}) + ||\mathbf{x} - \mathbf{y}|| \} \right] \right\}.$$

Therefore, the proof is completed using the definition of Moreau-Yosida regularization. $\square$

# D  DETAILS AND PROOFS IN SECTION 4.2

## D.1  DISCUSSIONS ON ONE-DIMENSIONAL CASE

The training for minimax game in GAN is difficult. When using traditional gradient methods, the training would suffer from some oscillatory behaviors (Goodfellow, 2017; Liang & Stokes, 2019). In order to better understand the optimization behaviors, we first study a one-dimension linear system that provides some insights on this problem. Such toy example (or a similar one) is also utilized by (Gidel et al., 2019; Nagarajan & Kolter, 2017) to shed lights on the instability of WGAN training[2]. Consider a linear critic $D_\psi(x) = \psi x$ and generator $G_\theta(z) = \theta z$. Then the Wasserstein GAN objective can be written as a constrained bilinear problem: $\min_\theta \max_{|\psi| \le 1} \psi \mathbb{E}[x] - \psi \theta \mathbb{E}[z]$, which could be further simplified as an unconstrained version (the behaviors can be generalized to multi-dimensional cases (Gidel et al., 2019)):

$$\min_\theta \max_\psi \psi - \psi \cdot \theta. \tag{9}$$

Unfortunately, such simple objective cannot guarantee convergence by traditional gradient methods like SGD with alternate updating[3]: $\theta_{k+1} = \theta_k + \eta \psi_k,, \psi_{k+1} = \psi_k + \eta(1 - \theta_{k+1})$. Such optimization would suffer from an oscillatory behavior, i.e., the updated parameters go around the optimum point ($[\psi^*, \theta^*] = [0, 1]$) forming a circle without converging to the centrality, which is shown in Fig. 3(a). A recent study in (Liang & Stokes, 2019) theoretically show that such oscillation is due to the interaction term in (9).

One solution to the instability of GAN training is to add (likelihood) regularization, which has been widely studied by recent literatures (Warde-Farley & Bengio, 2017; Li & Turner, 2018). With regularization term, the objective changes into $\min_\theta \max_{|\psi| \le 1} \psi \mathbb{E}[x] - \psi \theta \mathbb{E}[z] - \lambda \mathbb{E}[\log \mu(\theta z)]$, where $\mu(\cdot)$ denotes the likelihood function and $\lambda$ is a hyperparameter. A recent study (Tao et al., 2019) proves that when $\lambda < 0$ (likelihood-regularization), the extra term is equivalent to maximizing sample evidence, helping to stabilize GAN training; when $\lambda > 0$ (entropy-regularization), the extra term maximizes sample entropy, which encourages diversity of generator. Here we consider a Gaussian likelihood function for generated sample $x'$, $\mu(x') = \exp(-\frac{1}{2}(x' - b)^2)$ which is up to a constant. Its parameter can be estimated by $b = \mathbb{E}[x]$. Then for generated sample $x' = \theta z$, we have $\mathbb{E}(\log \mu(\theta z)) = -\frac{1}{2}\mathbb{E}[z^2]\theta^2 + \mathbb{E}[z]\mathbb{E}[x]\theta - \frac{1}{2}\mathbb{E}[x]^2$. Like the case in WGAN, we consider $\mathbb{E}[x] = \mathbb{E}[z] = 1$. Assume $\text{Var}[z] = 1$ and we have $\mathbb{E}[z^2] = 1 + \mathbb{E}[z]$. Hence, for the analysis on likelihood- (and entropy-) regularized WGAN, we can study the following system:

$$\min_\theta \max_\psi \psi - \psi \cdot \theta - \lambda(\theta^2 - \theta). \tag{10}$$

When $\lambda = 1$, the above objective degrades to (9); when $\lambda < 0$ (likelihood-regularization), the the gradient of regularization term pushes $\theta$ to shrink, which helps for convergence; when $\lambda > 0$ (entropy-regularization), the added term forms an amplifiying strength on $\theta$ and leads to divergence. Another issue of likelihood-regularization is that the extra term changes the optimum point and makes the model converge to a biased distribution, as proved by (Tao et al., 2019). In this case, one can verify that the optimum point becomes $[\psi^*, \theta^*] = [-\lambda, 1]$, resulting in a bias. To avoid this issue, (Tao et al., 2019) proposes to temporally decrease $|\lambda|$ through training. However, such method would also be stuck in oscillation when $|\lambda|$ gets close to zero as is shown in Fig. 3(a).

Finally, consider our proposed model. We also simplify the density estimator as a basic energy model $p_\phi(x) = \exp(-\frac{1}{2}x^2 - \phi x)$ whose score function $\nabla_x \log p_\phi(x) = -x - \phi$. Then

---

[2]Our theoretical discussions focus on WGAN, and we also compare with original GAN in the experiments.

[3]Here, we adopt the most widely used alternate updating strategy. The simultaneous updating, i.e., $\theta_{k+1} = \theta_k + \eta \psi_k$ and $\psi_{k+1} = \psi_k + \eta(1 - \theta_k)$, would diverge in this case.

if we specify the two Stein discrepancies in (3) as KSD with kernel $k(x_1, x_2) = \mathbb{I}(x_1 = x_2)$, then $\mathcal{S}(\mathbb{P}_{real}, \mathbb{P}_E) = \mathbb{E}_{x_1, x_2}[(\nabla_{x_1} \log p_\phi(x_1) - \nabla_{x_1} \log \mu(x_1))k(x_1, x_2)(\nabla_{x_2} \log p_\phi(x_2) - \nabla_{x_2} \log \mu(x_2))] = \mathbb{E}_x[(\nabla_x \log p_\phi(x) - \nabla_x \log \mu(x))^2] = (\phi + \mathbb{E}[x])^2$. Similarly, one can obtain $\mathcal{S}(\mathbb{P}_G, \mathbb{P}_E) = (\phi + \theta\mathbb{E}[z])^2$. Therefore we arrive at the objective in (11)

$$\min_\theta \max_\psi \min_\phi \psi - \psi \cdot \theta + \frac{\lambda_1}{2}(1+\phi)^2 + \frac{\lambda_2}{2}(\theta + \phi)^2. \tag{11}$$

Interestingly, for $\forall \lambda_1, \lambda_2$, the optimum remains the same $[\psi^*, \theta^*, \phi^*] = [0, 1, -1]$. Then we show that the optimization guarantees convergence to $[\psi^*, \theta^*, \phi^*]$.

**Proposition 1.** *Using alternate SGD for (11) geometrically decreases the square norm $N_t = |\psi^t|^2 + |\theta - 1|^2 + |\phi + 1|^2$, for any $0 < \eta < 1$ with $\lambda_1 = \lambda_2 = 1$,*

$$N_{t+1} = (1 - \eta^2(1-\eta)^2)N_t. \tag{12}$$

*Proof.* Instead of directly studying the optimization for (11), we first prove the following problem will converge to the unique optimum,

$$\min_\theta \max_\psi \min_\phi \theta\psi + \theta\phi + \frac{1}{2}\theta^2 + \phi^2. \tag{13}$$

Applying alternate SGD we have the following iterations:

$$\psi_{t+1} = \psi_t + \eta * \theta_t,$$
$$\phi_{t+1} = \phi_t - \eta * (\theta_t + 2\phi_t) = (1 - 2\eta)\phi_t - \eta\theta_t,$$
$$\theta_{t+1} = \theta_t - \eta(\psi_{t+1} + \phi_{t+1} + \theta_t) = -\eta(1 - 2\eta)\phi_t + (1 - \eta)\theta_t - \eta\psi_t.$$

Then we obtain the relationship between adjacent iterations:

$$\begin{bmatrix} \psi_{t+1} \\ \phi_{t+1} \\ \theta_{t+1} \end{bmatrix} = \begin{bmatrix} 1 & 0 & \eta \\ 0 & 1 - 2\eta & -\eta \\ -\eta & -\eta(1 - 2\eta) & 1 - \eta \end{bmatrix} \cdot \begin{bmatrix} \psi_t \\ \phi_t \\ \theta_t \end{bmatrix} = M \cdot \begin{bmatrix} \psi_t \\ \phi_t \\ \theta_t \end{bmatrix}$$

We further calculate the eigenvalues for matrix $M$ and have the following equations (assume the eigenvalue as $\lambda$):

$$(\lambda - 1)^3 + 3\eta(\lambda - 1)^2 + 2\eta^2(1 + \eta)(\lambda - 1) + 2\eta^3 = 0.$$

One can verify that the solutions to the above equation satisfy $|\lambda| < \sqrt{(1 - \eta + \eta^2)(1 + \eta - \eta^2)}$.

Then we have the following relationship

$$\left\| \begin{bmatrix} \psi_{t+1} \\ \phi_{t+1} \\ \theta_{t+1} \end{bmatrix} \right\|_2^2 = \left\| [\psi_t \quad \phi_t \quad \theta_t] \cdot M^\top M \cdot \begin{bmatrix} \psi_t \\ \phi_t \\ \theta_t \end{bmatrix} \right\|_2^2 \leq \lambda_m^2 \cdot \left\| \begin{bmatrix} \psi_t \\ \phi_t \\ \theta_t \end{bmatrix} \right\|_2^2$$

where $\lambda_m$ denotes the eigenvalue with the maximum absolute value of matrix $M$. Hence, we have

$$\psi_{t+1}^2 + \phi_{t+1}^2 + \theta_{t+1}^2 \leq (1 - \eta + \eta^2)(1 + \eta - \eta^2)[\psi_t^2 + \phi_t^2 + \theta_t^2].$$

We proceed to replace $\psi$, $\phi$ and $\theta$ in (13) by $\psi'$, $\phi'$ and $\theta'$ respectively and conduct a change of variable: let $\theta' = 1 - \theta$ and $\phi' = -1 - \phi$. Then we get the conclusion in the proposition.

$\square$

As shown in Fig. 3(a), Stein Bridging achieves a good convergence to the right optimum. Compared with (9), the objective (11) adds a new bilinear term $\phi \cdot \theta$, which acts like a connection between the generator and estimator, and two other quadratic terms, which help to penalize the increasing of values through training. The added terms and original terms in (11) cooperate to guarantee convergence to a unique optimum. In fact, the added terms $\frac{\lambda_1}{2}(1+\phi)^2 + \frac{\lambda_2}{2}(\theta+\phi)^2$ in (11) and the original terms $\psi - \psi \cdot \theta$ in WGAN play both necessary roles to guarantee the convergence to the unique optimum points $[\psi^*, \theta^*, \phi^*] = [0, 1, -1]$. If we remove the critic and optimize $\theta$ and $\phi$ with the remaining loss terms, we would find that the training would converge but not necessarily to $[\psi^*, \theta^*] = [0, 1]$ (since the optimum points are not unique in this case). On the other hand, if we remove the estimator, the system degrades to (9) and would not converge to the unique optimum point $[\psi^*, \theta^*] = [0, 1]$. If we consider both of the world and optimize three terms together, the training would converge to a unique global optimum $[\psi^*, \theta^*, \phi^*] = [0, 1, -1]$.

### D.2 GENERALIZATION TO BILINEAR SYSTEMS

Our analysis in the one-dimension case inspires us that we can add affiliated variable to modify the objective and stabilize the training for general bilinear system. The bilinear system is of wide interest for researchers focusing on stability of GAN training ((Goodfellow, 2017; Liang & Stokes, 2019; Gidel et al., 2019; Gemp & Mahadevan, 2018; Zhang & Yu, 2020)). The general bilinear function can be written as

$$F(\boldsymbol{\psi}, \boldsymbol{\theta}) = \boldsymbol{\theta}^\top \mathbf{A} \boldsymbol{\psi} - \mathbf{b}^\top \boldsymbol{\theta} - \mathbf{c}^\top \boldsymbol{\psi}, \tag{14}$$

where $\boldsymbol{\psi}, \boldsymbol{\theta}$ are both $r$-dimensional vectors and the objective is $\min_{\boldsymbol{\theta}} \max_{\boldsymbol{\psi}} F(\boldsymbol{\psi}, \boldsymbol{\theta})$ which can be seen as a basic form of various GAN objectives. Unfortunately, if we directly use simultaneous (resp. alternate) SGD to optimize such objectives, one can obtain divergence (resp. fluctuation). To solve the issue, some recent papers propose several optimization algorithms, like extrapolation from the past ((Gidel et al., 2019)), crossing the curl ((Gemp & Mahadevan, 2018)) and consensus optimization ((Liang & Stokes, 2019)). Also, (Liang & Stokes, 2019) shows that it is the interaction term which generates non-zero values for $\nabla_{\boldsymbol{\theta}\boldsymbol{\psi}} F$ and $\nabla_{\boldsymbol{\psi}\boldsymbol{\theta}} F$ that leads to such instability of training. Different from previous works that focused on algorithmic perspective, we propose to add new affiliated variables which modify the objective function and allow the SGD algorithm to achieve convergence without changing the optimum points.

Based on the minimax objective of (14) we add affiliated $r$-dimensional variable $\boldsymbol{\phi}$ (corresponding to the estimator in our model) the original system and tackle the following problem:

$$\min_{\boldsymbol{\theta}} \max_{\boldsymbol{\psi}} \min_{\boldsymbol{\phi}} F(\boldsymbol{\psi}, \boldsymbol{\theta}) + \alpha H(\boldsymbol{\phi}, \boldsymbol{\theta}), \tag{15}$$

where $H(\boldsymbol{\phi}, \boldsymbol{\theta}) = \frac{1}{2}(\boldsymbol{\theta} + \boldsymbol{\phi})^\top \mathbf{B}(\boldsymbol{\theta} + \boldsymbol{\phi})$, $\mathbf{B} = (\mathbf{A}\mathbf{A}^\top)^{\frac{1}{2}}$ and $\alpha$ is a non-negative constant. Theoretically, the new problem keeps the optimum points of (14) unchanged. Let $L(\boldsymbol{\psi}, \boldsymbol{\phi}, \boldsymbol{\theta}) = F(\boldsymbol{\psi}, \boldsymbol{\theta}) + \alpha G(\boldsymbol{\phi}, \boldsymbol{\theta})$.

**Proposition 2.** *Assume the optimum point of* $\min_{\boldsymbol{\theta}} \max_{\boldsymbol{\psi}} F(\boldsymbol{\psi}, \boldsymbol{\theta})$ *are* $[\boldsymbol{\psi}^*, \boldsymbol{\theta}^*]$, *then the optimum points of (15) would be* $[\boldsymbol{\psi}^*, \boldsymbol{\theta}^*, \boldsymbol{\phi}^*]$ *where* $\boldsymbol{\phi}^* = -\boldsymbol{\theta}^*$.

*Proof.* The condition tells us that $\nabla_{\boldsymbol{\theta}} F(\boldsymbol{\psi}^*, \boldsymbol{\theta}) = 0$ and $\nabla_{\boldsymbol{\psi}} F(\boldsymbol{\psi}, \boldsymbol{\theta}^*) = 0$. Then we derive the gradients for $L(\boldsymbol{\psi}, \boldsymbol{\phi}, \boldsymbol{\theta})$,

$$\nabla_{\boldsymbol{\psi}} L(\boldsymbol{\psi}^*, \boldsymbol{\phi}, \boldsymbol{\theta}) = \nabla_{\boldsymbol{\theta}} F(\boldsymbol{\psi}^*, \boldsymbol{\theta}) = 0, \tag{16}$$

$$\nabla_{\boldsymbol{\theta}} L(\boldsymbol{\psi}, \boldsymbol{\phi}, \boldsymbol{\theta}^*) = \nabla_{\boldsymbol{\theta}} F(\boldsymbol{\psi}, \boldsymbol{\theta}^*) + \nabla_{\boldsymbol{\theta}} H(\boldsymbol{\phi}, \boldsymbol{\theta}^*) = \frac{1}{2}(\mathbf{B} + \mathbf{B}^\top)(\boldsymbol{\theta}^* + \boldsymbol{\phi}), \tag{17}$$

$$\nabla_{\boldsymbol{\phi}} L(\boldsymbol{\psi}, \boldsymbol{\phi}, \boldsymbol{\theta}) = \nabla_{\boldsymbol{\phi}} H(\boldsymbol{\phi}, \boldsymbol{\theta}) = \frac{1}{2}(\mathbf{B} + \mathbf{B}^\top)(\boldsymbol{\phi} + \boldsymbol{\theta}), \tag{18}$$

Combining (17) and (18) we get $\boldsymbol{\phi}^* = -\boldsymbol{\theta}^*$. Hence, the optimum point of (15) is $[\boldsymbol{\psi}^*, \boldsymbol{\theta}^*, \boldsymbol{\phi}^*]$ where $\boldsymbol{\phi}^* = -\boldsymbol{\theta}^*$. $\qquad\square$

The advantage of the new problem is that it can be solved by SGD algorithm and guarantees convergence theoretically. We formulate the results in the following theorem.

**Theorem 3.** *For problem* $\min_{\boldsymbol{\theta}} \max_{\boldsymbol{\psi}} \min_{\boldsymbol{\phi}} L(\boldsymbol{\psi}, \boldsymbol{\phi}, \boldsymbol{\theta})$ *using alternate SGD algorithm, i.e.,*

$$\begin{aligned}
\boldsymbol{\psi}_{t+1} &= \boldsymbol{\psi}_t + \eta \nabla_{\boldsymbol{\psi}} L(\boldsymbol{\theta}_t, \boldsymbol{\psi}_t, \boldsymbol{\phi}_t), \\
\boldsymbol{\phi}_{t+1} &= \boldsymbol{\phi}_t - \eta \nabla_{\boldsymbol{\phi}} L(\boldsymbol{\theta}_t, \boldsymbol{\psi}_{t+1}, \boldsymbol{\phi}_t), \\
\boldsymbol{\theta}_{t+1} &= \boldsymbol{\theta}_t - \eta \nabla_{\boldsymbol{\theta}} L(\boldsymbol{\theta}_t, \boldsymbol{\psi}_{t+1}, \boldsymbol{\phi}_{t+1}),
\end{aligned} \tag{19}$$

*we can achieve convergence to* $[\boldsymbol{\psi}^*, \boldsymbol{\theta}^*, \boldsymbol{\phi}^*]$ *where* $\boldsymbol{\phi}^* = -\boldsymbol{\theta}^*$ *with at least linear rate of* $(1 - \eta_1 + \eta_2^2)(1 + \eta_2 - \eta_1^2)$ *where* $\eta_1 = \eta \sigma_{min}$, $\eta_2 = \eta \sigma_{max}$ *and* $\sigma_{min}$ *(resp.* $\sigma_{max}$*) denotes the maximum (resp. minimum) singular value of matrix* $\mathbf{A}$.

To prove Theorem 3, we can prove a more general argument.

**Lemma 1.** *If we consider any first-order optimization method on (15), i.e.,*

$$\psi_{t+1} \in \psi_0 + span(L(\psi_0, \phi, \theta), \cdots, F(\psi_t, \phi, \theta)), \forall t \in \mathbb{N},$$
$$\phi_{t+1} \in \psi_0 + span(L(\psi, \phi_0, \theta), \cdots, L(\psi, \phi_t, \theta)), \forall t \in \mathbb{N},$$
$$\theta_{t+1} \in \psi_0 + span(L(\psi, \phi, \theta_0), \cdots, L(\psi, \phi, \theta_t)), \forall t \in \mathbb{N},$$

*Then we have*

$$\widetilde{\psi}_t = \mathbf{V}^\top(\psi_t - \psi^*), \quad \widetilde{\phi}_t = \mathbf{U}^\top(\phi_t - \phi^*), \quad \widetilde{\theta}_t = \mathbf{U}^\top(\theta_t - \theta^*),$$

*where* $\mathbf{U}$ *and* $\mathbf{V}$ *are the singular vectors decomposed by matrix* $\mathbf{A}$ *using SVD decomposition, i.e.,* $\mathbf{A} = \mathbf{UDV}^\top$ *and the triple* $([\widetilde{\psi}_t]_i, [\widetilde{\phi}_t]_i, [\widetilde{\theta}_t]_i)_{1 \le i \le r}$ *follows the update rule with step size* $\sigma_i \eta$ *as the same optimization method on a unidimensional problem*

$$\min_\theta \max_\psi \min_\phi \theta\psi + \theta\phi + \frac{1}{2}\theta^2 + \frac{1}{2}\phi^2, \tag{20}$$

*with step size* $\eta$*, where* $\sigma_i$ *denotes the* $i$*-th singular value on the diagonal of* $\mathbf{D}$*.*

*Proof.* The proof is extended from the proof of Lemma 3 in (Gidel et al., 2019). The general class of first-order optimization methods derive the following updations:

$$\psi_{t+1} = \psi_0 + \sum_{s=0}^{t+1} \rho_{st}(\mathbf{A}^\top\theta_s - \mathbf{c}) = \psi_0 + \sum_{s=0}^{t+1} \rho_{st}\mathbf{A}^\top(\theta_s - \theta^*),$$

$$\phi_{t+1} = \phi_0 + \frac{1}{2}\sum_{s=0}^{t+1} \delta_{st}(\mathbf{B} + \mathbf{B}^\top)(\theta_s + \phi_s),$$

$$\theta_{t+1} = \theta_0 + \sum_{s=0}^{t+1} \mu_{st}[\mathbf{A}(\psi_s - \psi^*) + \frac{1}{2}(\mathbf{B} + \mathbf{B}^\top)(\theta_s + \phi_s)],$$

where $\rho_{st}, \delta_{st}, \mu_{st} \in \mathbb{R}$ depend on specific optimization method (for example, in SGD, $\rho_{tt} = \delta_{tt} = \mu_{tt}$ remain as a non-zero constant for $\forall t$ and other coefficients are zero).

Using SVD $\mathbf{A} = \mathbf{UDV}^\top$ and the fact $\theta^* = -\phi^*$, $\mathbf{B} = (\mathbf{UDD}^\top\mathbf{U}^\top) = \mathbf{D}$, we have

$$\mathbf{V}^\top(\psi_{t+1} - \psi^*) = \mathbf{V}^\top(\psi_0 - \psi^*) + \sum_{s=0}^{t+1} \rho_{st}\mathbf{D}^\top\mathbf{U}^\top(\theta_s - \theta^*)$$

$$\mathbf{U}^\top(\phi_{t+1} - \phi^*) = \mathbf{U}^\top(\phi_0 - \phi^*) + \sum_{s=0}^{t+1} \delta_{st}\mathbf{U}^\top\mathbf{D}(\theta_s - \theta^*) + \mathbf{U}^\top\mathbf{D}(\phi_s - \phi^*),$$

$$\mathbf{U}^\top(\theta_{t+1} - \theta^*) = \mathbf{U}^\top(\theta_0 - \theta^*) + \sum_{s=0}^{t+1} \rho_{st}[\mathbf{DV}^\top(\psi_s - \psi^*) + \mathbf{U}^\top\mathbf{D}(\theta_s - \theta^*) + \mathbf{U}^\top\mathbf{D}(\phi_s - \phi^*)],$$

and equivalently,

$$\widetilde{\psi}_{t+1} = \widetilde{\psi}_0 + \sum_{s=0}^{t+1} \rho_{st}\mathbf{D}^\top\widetilde{\theta}_t, \quad \widetilde{\phi}_t = \widetilde{\phi}_0 + \sum_{s=0}^{t+1} \delta_{st}\mathbf{D}(\widetilde{\theta}_t + \widetilde{\phi}_t),$$

$$\widetilde{\theta}_{t+1} = \widetilde{\theta}_0 + \sum_{s=0}^{t+1} \rho_{st}\mathbf{D}(\widetilde{\psi}_t + \widetilde{\theta}_t + \widetilde{\phi}_t).$$

Note that $\mathbf{D}$ is a rectangular matrix with non-zero elements on a diagonal block of size $r$. Hence, the above $r$-dimensional problem can be reduced to $r$ unidimensional problems:

$$[\widetilde{\psi}_{t+1}]_i = [\widetilde{\psi}_0]_i + \sum_{s=0}^{t+1} \rho_{st}\sigma_i[\widetilde{\theta}_t]_i, \quad [\widetilde{\phi}_t]_i = [\widetilde{\phi}_0]_i + \sum_{s=0}^{t+1} \delta_{st}\sigma_i([\widetilde{\theta}_t]_i + [\widetilde{\phi}_t]_i),$$

$$[\widetilde{\theta}_{t+1}]_i = [\widetilde{\theta}_0]_i + \sum_{s=0}^{t+1} \rho_{st}\sigma_i([\widetilde{\psi}_t]_i + [\widetilde{\theta}_t]_i + [\widetilde{\phi}_t]_i).$$

The above iterations can be conducted independently in each dimension where the optimization in $i$-th dimension follows the same updating rule with step size $\sigma_i \eta$ as problem in (20). $\qquad\square$

Furthermore, since problem (20) can achieve convergence with a linear rate of $(1-\eta+\eta^2)(1+\eta-\eta^2)$ using alternate SGD (the proof is similar to that of (13)), the multi-dimensional problem in (15) can achieve convergence by SGD with at least a rate of $(1 - \eta_1 + \eta_2^2)(1 + \eta_2 - \eta_1^2)$ where $\eta_1 = \eta\sigma_{max}$, $\eta_2 = \eta\sigma_{min}$ and $\sigma_{max}$ (resp. $\sigma_{min}$) denotes the maximum (resp. minimum) singular value of matrix $\mathbf{A}$. We conclude the proof for Theorem 4.

Theorem 3 suggests that the added term $H(\boldsymbol{\phi}, \boldsymbol{\theta})$ with affiliated variables $\phi$ could help the SGD algorithm achieve convergence to the the same optimum points as directly optimizing $F(\boldsymbol{\psi}, \boldsymbol{\theta})$. Our method is related to consensus optimization algorithm ((Liang & Stokes, 2019)) which adds a regularization term $\|\nabla_{\boldsymbol{\theta}} F(\boldsymbol{\psi}, \boldsymbol{\theta})\| + \|\nabla_{\boldsymbol{\psi}} F(\boldsymbol{\psi}, \boldsymbol{\theta})\|$ to (14) resulting extra quadratic terms for $\boldsymbol{\theta}$ and $\boldsymbol{\psi}$. The disadvantage of such method is the requirement of Hessian matrix of $F(\boldsymbol{\psi}, \boldsymbol{\theta})$ which is computational expensive for high-dimensional data. By contrast, our solution only requires the first-order derivatives.

# E DETAILS FOR IMPLEMENTATIONS

## E.1 SYNTHETIC DATASETS

We provide the details for two synthetic datasets. The Two-Circle dataset consists of 24 Gaussian mixtures where 8 of them are located in an inner circle with radius $r_1 = 4$ and 16 of them lie in an outer circle with radius $r_2 = 8$. For each Gaussian component, the covariance matrix is $\begin{pmatrix} 0.2 & 0 \\ 0 & 0.2 \end{pmatrix} = \sigma_1 \mathbf{I}$ and the mean value is $[r_1 \cos t, r_1 \sin t]$, where $t = \frac{2\pi \cdot k}{8}$, $k = 1, \cdots, 8$, for the inner circle, and $[r_2 \cos t, r_2 \sin t]$, where $t = \frac{2\pi \cdot k}{16}$, $k = 1, \cdots, 16$ for the outer circle. We sample $N_1 = 2000$ points as true observed samples for model training.

The Two-Spiral dataset contains 100 Gaussian mixtures whose centers locate on two spiral-shaped curves. For each Gaussian component, the covariance matrix is $\begin{pmatrix} 0.5 & 0 \\ 0 & 0.5 \end{pmatrix} = \sigma_2 \mathbf{I}$ and the mean value is $[-c_1 \cos c_1, c_1 \sin c_1]$, where $c_1 = \frac{2\pi}{3} + linspace(0, 0.5, 50) \cdot 2\pi$, for one spiral, and $[c_2 \cos c_2, -c_2 \sin c_2]$, where $c_2 = \frac{2\pi}{3} + linspace(0, 0.5, 50) \cdot 2\pi$ for another spiral. We sample $N_2 = 5000$ points as true observed samples.

## E.2 MODEL SPECIFICATIONS AND TRAINING ALGORITHM

In different tasks, we consider different model specifications in order to meet the demand of capacify as well as test the effectiveness under various settings. Our proposed framework (3) adopts Wasserstein distance for the first term and two Stein discrepancies for the second and the third terms. We can write (3) as a more general form

$$\min_{\theta, \phi} \mathcal{D}_1(\mathbb{P}_{\text{real}}, \mathbb{P}_G) + \lambda_1 \mathcal{D}_2(\mathbb{P}_{\text{real}}, \mathbb{P}_E) + \lambda_2 \mathcal{D}_3(\mathbb{P}_G, \mathbb{P}_E), \tag{21}$$

where $\mathcal{D}_1$, $\mathcal{D}_2$, $\mathcal{D}_3$ denote three general discrepancy measures for distributions. As stated in our remark, $\mathcal{D}_1$ can be specified as arbitrary discrepancy measures for implicit generative models. Here we also use JS divergence, the objective for valina GAN. To well distinguish them, we call the model using Wasserstein distance (resp. JS divergence) as Joint-W (resp. Joint-JS) in our experiments. On the other hand, the two Stein discrepancies in (3) can be specified by KSD (as defined by $\mathcal{S}_k$ in (5)) or general Stein discrepancy with an extra critic (as defined by $\mathcal{S}$ in (1)). Hence, the two specifications for $\mathcal{D}_1$ and the two for $\mathcal{D}_2$ ($\mathcal{D}_3$) compose four different combinations in total, and we organize the objectives in each case in Table 3.

In our experiments, we use KSD with RBF kernels for $\mathcal{D}_2$ and $\mathcal{D}_3$ in Joint-W and Joint-JS on two synthetic datasets. For MNIST with conditional training (given the digit class as model input), we also use KSD with RBF kernels. For MNIST and CIFAR with unconditional training (the class is not given as known information), we find that KSD cannot provide desirable results so we adopt general Stein discrepancy for higher model capacity.

The objectives in Table 3 appear to be comutationally expensive. In the worst case (using general Stein discrepancy), there are two minimax operations where one is from GAN or WGAN and one is from Stein discrepancy estimation. To guarantee training efficiency, we alternatively update the

Table 3: Objectives for different specifications of $\mathcal{D}_1(\mathbb{P}_{\text{real}}, \mathbb{P}_G)$, $\mathcal{D}_2(\mathbb{P}_{\text{real}}, \mathbb{P}_E)$ and $\mathcal{D}_3(\mathbb{P}_G, \mathbb{P}_E)$. We specify $\mathcal{D}_1$ as Wasserstein distance or JS divergence in our paper and for $\mathcal{D}_2$ and $\mathcal{D}_3$ we consider the general Stein discrepancy or kernel Stein discrepancy. Here we use $\mathcal{W}$, $\mathcal{JS}$ to denote Wasserstein distance and JS divergence respectively, and $\mathcal{S}$, $\mathcal{S}_k$ to represent general Stein discrepancy and kernel Stein discrepancy respectively. We omit the gradient penalty term for Wasserstein distance here but use it in experiments.

| $\mathcal{D}_1$ | $\mathcal{D}_2$ | $\mathcal{D}_3$ | Objective |
|---|---|---|---|
| $\mathcal{W}$ | $\mathcal{S}$ | $\mathcal{S}$ | $\min_\theta \min_\phi \max_\psi \max_\pi \mathbb{E}_{\mathbf{x}\sim\mathbb{P}_{data}}[d_\psi(\mathbf{x})] - \mathbb{E}_{\mathbf{z}\sim p_0}[d_\psi(G_\theta(\mathbf{z}))]$ $+ \lambda_1 \mathbb{E}_{\mathbf{x}\sim\mathbb{P}_{data}}[\mathcal{A}_{p_\phi}[\mathbf{f}_\pi(\mathbf{x})]] + \lambda_2 \mathbb{E}_{\mathbf{z}\sim\wr_0}[\mathcal{A}_{p_\phi}[\mathbf{f}_\pi(G_\theta(\mathbf{z}))]]$ |
| $\mathcal{W}$ | $\mathcal{S}_k$ | $\mathcal{S}_k$ | $\min_\theta \min_\phi \max_\psi \mathbb{E}_{\mathbf{x}\sim\mathbb{P}_{data}}[d_\psi(\mathbf{x})] - \mathbb{E}_{\mathbf{z}\sim p_0}[d_\psi(G_\theta(\mathbf{z}))]$ $+ \lambda_1 \mathbb{E}_{\mathbf{x},\mathbf{x}'\sim\mathbb{P}_{data}}[u_{p_\phi}(x,x')] + \lambda_2 \mathbb{E}_{\mathbf{z},\mathbf{z}'\sim p_0}[u_{p_\phi}(G_\theta(\mathbf{z}), G_\theta(\mathbf{z}'))]$ |
| $\mathcal{JS}$ | $\mathcal{S}$ | $\mathcal{S}$ | $\min_\theta \min_\phi \max_\psi \max_\pi \mathbb{E}_{\mathbf{x}\sim\mathbb{P}_r}[\log(d_\psi(\mathbf{x}))] + \mathbb{E}_{\mathbf{z}\sim p_0}[\log(1 - d_\psi(G_\theta(\mathbf{z})))]$ $+ \lambda_1 \mathbb{E}_{\mathbf{x}\sim\mathbb{P}_{data}}[\mathcal{A}_{p_\phi}[\mathbf{f}_\pi(\mathbf{x})]] + \lambda_2 \mathbb{E}_{\mathbf{z}\sim\wr_0}[\mathcal{A}_{p_\phi}[\mathbf{f}_\pi(G_\theta(\mathbf{z}))]]$ |
| $\mathcal{JS}$ | $\mathcal{S}_k$ | $\mathcal{S}_k$ | $\min_\theta \min_\phi \max_\psi \mathbb{E}_{\mathbf{x}\sim\mathbb{P}_r}[\log(d_\psi(\mathbf{x}))] + \mathbb{E}_{\mathbf{z}\sim p_0}[\log(1 - d_\psi(G_\theta(\mathbf{z})))]$ $+ \lambda_1 \mathbb{E}_{\mathbf{x},\mathbf{x}'\sim\mathbb{P}_{data}}[u_{p_\phi}(x,x')] + \lambda_2 \mathbb{E}_{\mathbf{z},\mathbf{z}'\sim p_0}[u_{p_\phi}(G_\theta(\mathbf{z}), G_\theta(\mathbf{z}'))]$ |

generator, estimator, Wasserstein critic and Stein critic over the parameters $\theta$, $\phi$, $\psi$ and $\pi$ respectively. Specifically, in one iteration, we optimize the generator over $\theta$ and the estimator over $\phi$ with one step respectively, and then optimize the Wasserstein critic over $\psi$ with $n_d$ steps and the Stein critic over $\pi$ with $n_c$ steps. Such training approach guarantees the same time complexity order of proposed method as that of GAN or WGAN, and the training time for our model can be bounded within constant times the time for training GAN model. In our experiment, we set $n_d = n_c = 5$ and empirically find that our model Stein Bridging would be two times slower than WGAN on average. We present the training algorithm for Stein Bridging in Algorithm 1.

### E.3 IMPLEMENTATION DETAILS

We give the information of network architectures and hyper-parameter settings for our model as well as each competitor in our experiments.

The energy function is often parametrized as a sum of multiple experts ((Hinton, 1999)) and each expert can have various function forms depending on the distributions. If using sigmoid distribution, the energy function becomes (see section 2.1 in (Kim & Bengio, 2017) for details)

$$E_\phi(\mathbf{x}) = \sum_i \log(1 + e^{-(\mathbf{W}_i n(\mathbf{x}) + b_i)}), \tag{22}$$

where $n(\mathbf{x})$ maps input $\mathbf{x}$ to a feature vector and could be specified as a deep neural network, which corresponds to deep energy model ((Ngiam et al., 2011))

When not using KSD, the implementation for Stein critic $\mathbf{f}$ and operation function $\phi$ in (1) has still remained an open problem. Some existing studies like (Hu et al., 2018) set $d' = 1$ in which situation $\mathbf{f}$ reduces to a scalar-function from $d$-dimension input to one-dimension scalar value. Such setting can reduce computational cost since large $d'$ could lead to heavy computation for training. Empirically, in our experiments on image dataset, we find that setting $d' = 1$ can provide similar performance to $d' = 10$ or $d' = 100$. Hence, we set $d' = 1$ in our experiment in order for efficiency. Besides, to further reduce computational cost, we let the two Stein critics share the parameters, which empirically provide better performance than two different Stein critics.

Another tricky point is how to design a proper $\Gamma$ given $d' \neq d$ where the trace operation is not applicable. One simple way is to set $\Gamma$ as some matrix norms. However, the issue is that using matrix norm would make it hard for SGD learning. The reason is that the $\Gamma$ and the expectation in (1) cannot exchange the order, in which case there is no unbiased estimation by mini-batch samples for the gradient. Here, we specify $\Gamma$ as max-pooling over different dimensions of $\mathcal{A}_{p_\phi}[\mathbf{f}_\pi(\mathbf{x})]$, i.e. the gradient would back-propagate through the dimension with largest absolute value at one time. Theoretically, such setting can guarantee the value in each dimension reduces to zero through training and we find it works well in practice.

---

**Algorithm 1:** Training Algorithm for Stein Bridging

---

1    **REQUIRE:** observed training samples $\{\mathbf{x}\} \sim \mathbb{P}_{real}$.

2    **REQUIRE:** $\theta_0, \phi_0, \psi_0, \pi_0$, initial parameters for generator, estimator, Wasserstein critic and Stein critic models respectively. $\alpha^E = 0.0002, \beta_1^E = 0.9, \beta_2^E = 0.999$, Adam hyper-parameters for explicit models. $\alpha^I = 0.0002, \beta_1^I = 0.5, \beta_2^I = 0.999$, Adam hyper-parameters for implicit models. $\lambda_1 = 1, \lambda_2$, weights for $\mathcal{D}_2$ and $\mathcal{D}_3$ (we suggest increasing $\lambda_2$ from 0 to 1 through training). $n_d = 5, n_c = 5$ number of iterations for Wasserstein critic and Stein critic, respectively, before one iteration for generator and estimator. $B = 100$, batch size.

3    **while** *not converged* **do**

4      **for** $n = 1, \cdots, n_d$ **do**

5        Sample $B$ true samples $\{\mathbf{x}_i\}_{i=1}^B$ from $\{\mathbf{x}\}$;

6        Sample $B$ random noise $\{\mathbf{z}_i\}_{i=1}^B \sim P_0$ and obtain generated samples $\widetilde{\mathbf{x}}_{\mathbf{i}} = G_\theta(\mathbf{z}_i)$ ;

7        $\mathcal{L}_{dis} = \frac{1}{B}\sum_{i=1}^B d_\psi(\mathbf{x}_i) - d_\psi(\widetilde{\mathbf{x}}_{\mathbf{i}}) - \lambda(\|\nabla_{\hat{\mathbf{x}}_{\mathbf{i}}} d_\psi(\hat{\mathbf{x}}_{\mathbf{i}})\| - 1)^2$ // the last term is for gradient penalty in WGAN-GP where $\hat{\mathbf{x}}_{\mathbf{i}} = \epsilon_i \mathbf{x}_i + (1 - \epsilon_i)\widetilde{\mathbf{x}}_{\mathbf{i}}, \epsilon_i \sim U(0,1)$;

8        $\psi_{k+1} \leftarrow Adam(-\mathcal{L}_{dis}, \psi_k, \alpha^I, \beta_1^I, \beta_2^I)$// update the Wasserstein critic;

9      **for** $n = 1, \cdots, n_c$ **do**

10       Sample $B$ true samples $\{\mathbf{x}_i\}_{i=1}^B$ from $\{\mathbf{x}\}$;

11       Sample $B$ random noise $\{\mathbf{z}_i\}_{i=1}^B \sim P_0$ and obtain generated samples $\widetilde{\mathbf{x}}_{\mathbf{i}} = G_\theta(\mathbf{z}_i)$ ;

12       $\mathcal{L}_{critic} = \frac{1}{B}\sum_{i=1}^B \lambda_1 \mathcal{A}_{p_\phi}[\mathbf{f}_\pi(\mathbf{x})] + \lambda_2 \mathcal{A}_{p_\phi}[\mathbf{f}_\pi(\widetilde{\mathbf{x}}_{\mathbf{i}})]$;

13       $\pi_{k+1} \leftarrow Adam(-\mathcal{L}_{critic}, \pi_k, \alpha^E, \beta_1^E, \beta_2^E)$// update the Stein critic;

14      Sample $B$ random noise $\{\mathbf{z}_i\}_{i=1}^B \sim P_0$ and obtain generated samples $\widetilde{\mathbf{x}}_{\mathbf{i}} = G_\theta(\mathbf{z}_i)$ ;

15      $\mathcal{L}_{est} = \frac{1}{B}\sum_{i=1}^B \lambda_1 \mathcal{A}_{p_\phi}[\mathbf{f}_\pi(\mathbf{x})] + \lambda_2 \mathcal{A}_{p_\phi}[\mathbf{f}_\pi(\widetilde{\mathbf{x}}_{\mathbf{i}})]$;

16      $\phi_{k+1} \leftarrow Adam(\mathcal{L}_{est}, \phi_k, \alpha^E, \beta_1^E, \beta_2^E)$// update the density estimator;

17      $\mathcal{L}_{gen} = \frac{1}{B}\sum_{i=1}^B -d_\psi(\widetilde{\mathbf{x}}_{\mathbf{i}}) + \lambda_2 \mathcal{A}_{p_\phi}[\mathbf{f}_\pi(\widetilde{\mathbf{x}}_{\mathbf{i}})]$;

18      $\theta_{k+1} \leftarrow Adam(\mathcal{L}_{gen}, \theta_k, \alpha^I, \beta_1^I, \beta_2^I)$// update the sample generator;

19    **OUTPUT:** trained sample generator $G_\theta(\mathbf{z})$ and density estimator $p_\phi(\mathbf{x})$.

---

For synthetic datasets, we set the noise dimension as 4. All the generators are specified as a three-layer fully-connected (FC) neural network with neuron size $4-128-128-2$, and all the Wasserstein critics (or the discriminators in JS-divergence-based GAN) are also a three-layer FC network with neuron size $2-128-128-1$. For the estimators, we set the expert number as 4 and the feature function $n(\mathbf{x})$ is a FC network with neuron size $2-128-128-4$. Then in the last layer we sum the outputs from each expert as the energy value $E(\mathbf{x})$. The activation units are searched within $[LeakyReLU, tanh, sigmoid, softplus]$. The learning rate $[1e-6, 1e-5, 1e-4, 1e-3, 1e-2]$ and the batch size $[50, 100, 150, 200]$. The gradient penalty weight for WGAN is searched in $[0, 0.1, 1, 10, 100]$.

For MNIST dataset, we set the noise dimension as 100. All the critics/discriminators are implemented as a four-layer network where the first two layers adopt convolution operations with filter size 5 and stride $[2, 2]$ and the last two layers are FC layers. The size for each layer is $1-64-128-256-1$. All the generators are implemented as a four-layer networks where the first two layers are FC and the last two adopt deconvolution operations with filter size 5 and stride $[2, 2]$. The size for each layer is $100-256-128-64-1$. For the estimators, we consider the expert number as 128 and the feature function is the same as the Wasserstein critic except that the size of last layer is 128. Then we sum the outputs from each expert as the energy value. The activation units are searched within $[ReLU, LeakyReLU, tanh]$. The learning rate $[2e-5, 2e-4, 2e-3, 2e-2]$ and the batch size $[32, 64, 100, 128]$. The gradient penalty weight for WGAN is searched in $[1, 10, 100, 1000]$.

For CIFAR dataset, we adopt the same architecture as DCGAN for critics and generators. As for the estimator, the architecture of feature function is the same as the critics except the last year where we set the expert number as 128 and sum each output as the output energy value. The architectures for Stein critic are the same as Wasserstein critic for both MNIST and CIFAR datasets. In other words,

Table 4: Distances between means of generated digits (resp. images) and true digits (resp. images) on MNIST (resp. CIFAR-10).

|  | MNIST | | CIFAR | |
| --- | --- | --- | --- | --- |
| Method | $l_1$ Dis | $l_2$ Dis | $l_1$ Dis | $l_2$ Dis |
| WGAN-GP | 13.80 | 0.93 | 80.98 | 1.72 |
| WGAN+LR | 12.91 | 0.86 | 82.96 | 1.81 |
| WGAN+ER | 12.26 | 0.77 | 72.28 | 1.59 |
| WGAN+VA | 12.38 | 0.78 | 69.01 | 1.53 |
| DGM | 12.12 | 0.79 | 179.30 | 3.95 |
| Joint-W | **11.82** | **0.73** | **64.23** | **1.41** |

Table 5: Quantitative results including MMD (lower is better), HSR (higher is better) as the metrics for quality of generated samples and KLD (lower is better), JSD (lower is better), AUC (higher is better) as the metrics for accuracy of estimated densities on Two-Circle and Two-Spiral datasets.

|  | Two-Cirlce | | | | | Two-Spiral | | | | |
| --- | --- | --- | --- | --- | --- | --- | --- | --- | --- | --- |
| Method | MMD | HSR | KLD | JSD | AUC | MMD | HSR | KLD | JSD | AUC |
| GAN | 0.0033 | 0.772 | - | - | - | 0.0082 | 0.583 | - | - | - |
| GAN+VA | 0.0118 | 0.295 | - | - | - | 0.0085 | 0.761 | - | - | - |
| WGAN-GP | 0.0010 | 0.841 | - | - | - | 0.0090 | 0.697 | - | - | - |
| WGAN+VA | 0.0016 | 0.835 | - | - | - | 0.0159 | 0.618 | - | - | - |
| DEM | - | - | 2.036 | 0.431 | 0.683 | - | - | 1.206 | 0.315 | 0.640 |
| EGAN | - | - | 3.350 | 0.474 | 0.616 | - | - | 1.916 | 0.445 | 0.499 |
| DGM | 0.0040 | 0.774 | 2.272 | 0.445 | 0.600 | 0.0019 | 0.833 | 1.725 | 0.414 | 0.589 |
| Joint-JS | 0.0037 | **0.883** | 1.104 | 0.297 | **0.962** | 0.0031 | 0.717 | 0.655 | 0.193 | 0.808 |
| Joint-W | **0.0007** | 0.844 | **1.030** | **0.281** | 0.961 | **0.0003** | **0.909** | **0.364** | **0.110** | **0.810** |

we consider $d' = 1$ in (1) and further simply $\phi$ as an average of each dimension of $\mathbb{E}_{\mathbf{x} \sim \mathbb{P}}[\mathcal{A}_{\mathbb{Q}}\mathbf{f}(\mathbf{x})]$. Empirically we found this setting can provide efficient computation and decent performance.

### E.4 EVALUATION METRICS

We adopt some quantitative metrics to evaluate the performance of each method on different tasks. In section 4.1, we use two metrics to test the sample quality: Maximum Mean Discrepancy (MMD) and High-quality Sample Rate (HSR). MMD measures the discrepancy between two distributions $X$ and $Y$, $MMD(X, Y) = \|\frac{1}{n}\sum_{i=1}^{n}\Phi(x_i) - \frac{1}{m}\sum_{j=1}^{m}\Phi(y_i)\|$ where $x_i$ and $y_j$ denote samples from $X$ and $Y$ respectively and $\Phi$ maps each sample to a RKHS. Here we use RBF kernel and calculate MMD between generated samples and true samples. HSR statistics the rate of high-quality samples over all generated samples. For Two-Cirlce dataset, we define the generated points whose distance from the nearest Gaussian component is less than $\sigma_1$ as high-quality samples. We generate 2000 points in total and statistic HSR. For Two-Spiral dataset, we set the distance threshold as $5\sigma_2$ and generate 5000 points to calculate HSR. For CIFAR, we use the Inception V3 Network in Tensorflow as pre-trained classifier to calculate inception score.

In section 4.2, we use three metrics to characterize the performance for density estimation: KL divergence, JS divergence and AUC. We divide the map into a 300 meshgrid, calculate the unnormalized density values of each point given by the estimators and compute the KL and JS divergences between estimated density and ground-truth density. Besides, we select the centers of each Gaussian components as positive examples (expected to have high densities) and randomly sample 10 points within a circle around each center as negative examples (expected to have relatively low densities) and rank them according to the densities given by the model. Then we obtain the area under the curve (AUC) for false-positive rate v.s. true-positive rate.

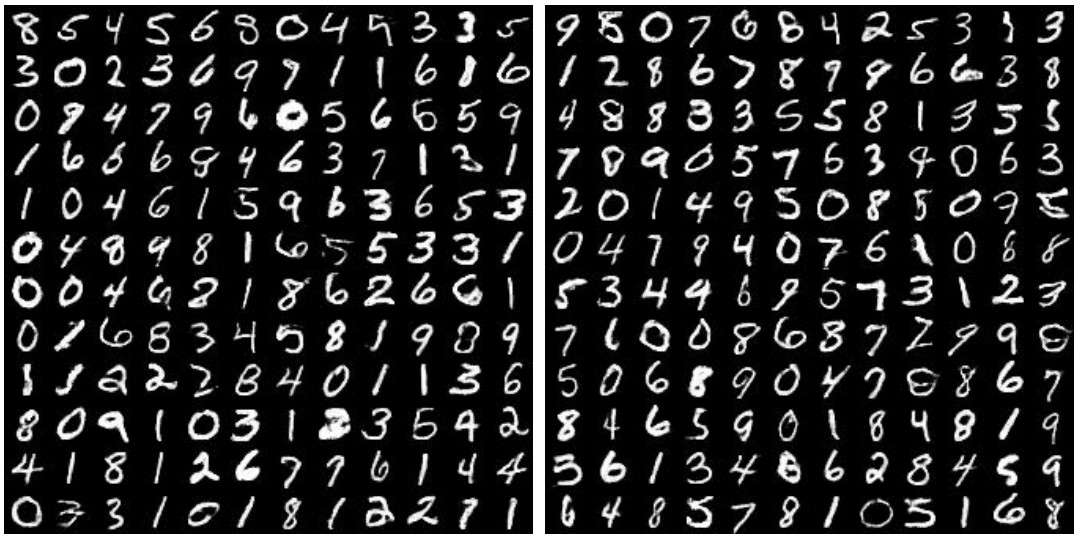

(a) Randomly sampled over all digits

(b) Randomly sampled over digits with top 50% densities

Figure 9: Generated digits given by Joint-W on MNIST.

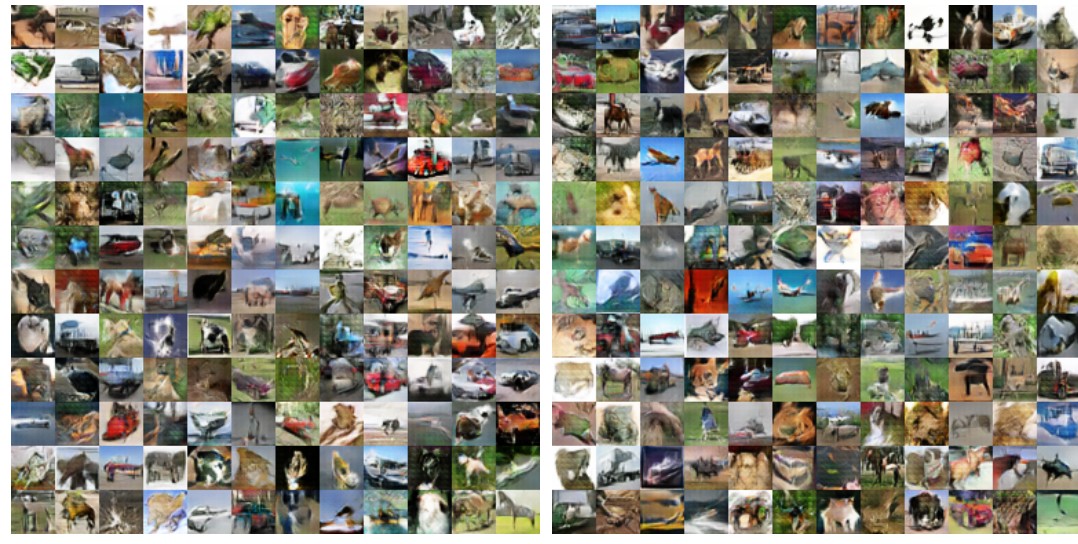

(a) Randomly sampled over all images

(b) Randomly sampled over images with top 50% densities

Figure 10: Generated images given by Joint-W on CIFAR.

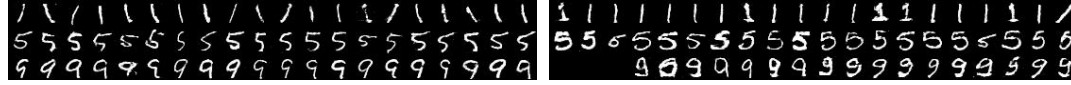

(a) Generated digits with highest densities          (b) Generated digits with lowest densities

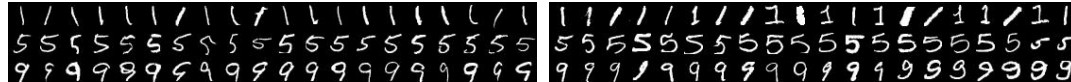

(c) Real digits with highest densities          (d) Real digits with lowest densities

Figure 11: The generated digits (and real digits) with the highest densities and the lowest densities given by Joint-W.

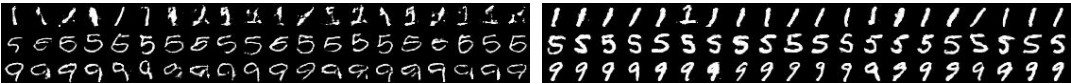

(a) Generated digits with highest densities          (b) Generated digits with lowest densities

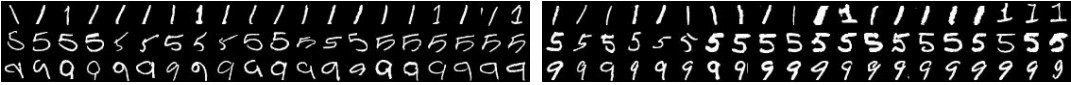

(c) Real digits with highest densities          (d) Real digits with lowest densities

Figure 12: The generated digits (and real digits) with the highest densities and the lowest densities given by DGM.

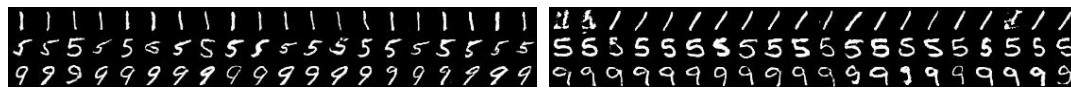

(a) Generated digits with highest densities          (b) Generated digits with lowest densities

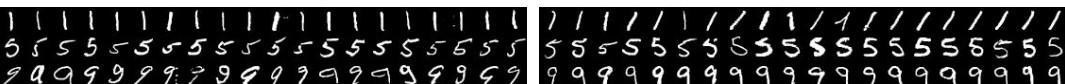

(c) Real digits with highest densities          (d) Real digits with lowest densities

Figure 13: The generated digits (and real digits) with the highest densities and the lowest densities given by EGAN.

