# OpenReview forum: "Mutual Calibration between Explicit and Implicit Deep Generative Models"
_ICLR.cc/2021/Conference — Reject_

### Official Review · AnonReviewer2 · 2020-10-27
**Interesting paper with some gaps in theoretical analysis**

**Rating:** 5
**Confidence:** 3

**Review:**

Summary of the paper:

The author proposed a novel regularization technique for jointly training of implicit model (IM) and explicit model (EM). This is achieved by connecting the training of IM and EM via Stein discrepancy (SD), which is called Stein bridging.

The author claimed this regularization can (i) smooth the Wasserstein critic by kernel Sobolev dual norm (ii) smooth the Stein critic by Moreau-Yoshida regularization (iii) stabilize the training of IM.

The author also theoretically proves the (i), (ii) under Wasserstein distance, and (iii) for a simple toy case.  The author empirically evaluated the EM by inspecting the mode coverage of toy example, ranking digit and OOD detection on more complex data sets.

-----
Reviews:

Clarity: The main text of this paper is in general clearly written and easy to follow. However, I have some concerns related to the theoretical analysis in the Appendix.

Novelty: This regularization technique seems to be novel to the best of my knowledge. Although the idea is simple, the author provided some theoretical analysis to back it up. However, it is not enough to fully back the claims made in the main text. Details later.

Technical soundness and concerns: I have some concerns related to the SD and proof for theorem 1 and 2.
1. For SD, the author mentioned that for Stein critic $f_s(\pmb{x})$, it is not necessary $\mathbb{R}^d\rightarrow \mathbb{R}^d$. One can specify a lower dimension $d'<d$ and make $f_s:\mathbb{R}^d\rightarrow \mathbb{R}^{d'}$ as long as $f_s$ belongs to the Stein class. Indeed, this is true for Stein's identity (see Def 2.1 in Liu's paper). However, this does not mean it defines a valid discrepancy measure. The original SD in Gorham's paper assumes $d'=d$ and the trace operator is used to transform $d\times d$ Stein identity to the scalar value. In that case, Gorham proves its validity by investigating its weak convergence property. My concern is I cannot see the direct generalization from the trace operator to other matrix operators like the ones used in this paper with $d'=1$. In other words, I agree that for two distributions $p$,$q$, when $p=q$, the SD defined in this paper is $0$, but not vice versa. Could the author point out any references or provides any details on the validity of the proposed SD?
2. I do not fully follow the derivations in Appendix C.1. In page 15, how do you introduce the auxiliary variable $r^2$? Why there are two $\min$ operations instead of one $\min$ with jointly optimizing $h$ and $r$? Could you elaborate more on this and also how do you get rid of $r^2$ in the constraints in the second equality?
3. I am also a bit confused about derivations in Appendix C.2. How do you combine the $\mathbb{P}$ and $\gamma$ in one $\min$ operation instead of $\min_{\mathbb{P}}\min_{\gamma}$? (In page 15)
4. In the main text, the author claimed that other objectives can be used for training implicit the model such as JS divergence. However, the theoretical analysis (theorem 1 and 2) only shows the regularization effect of Stein bridging is only for Wasserstein-1. So the analysis won't hold for other objectives like JS divergence.
5. It is known that Stein based divergence is a weak objective for training EBM (see Liu 2016). Therefore the regularization technique may help a lot, like the mode coverage demonstrated in the experiment. I wonder if SOTA training method for EBM is used (like SSM in Song 2020, Song 2019), does this regularization help the training, because this regularization is not cheap to compute (higher than the SOTA method for EBM).
6. In figure 4, I cannot find DEM and EGAN in the density plot.
7. In table 2, it seems that the training of DEM is failing as the AUC is closed to 0.5. Any guess on why it fails? How do you pre-process the data set? Do you add different scales of noise in the images to smooth it for the EBM to learn the distribution (like the trick used in Song 2019)?

Summary:
I am quite interested in this approach. But I am a bit concerned about the theoretical analysis and the true advantage of training EBM with Stein bridging compared to the cheaper SOTA EBM method.
--------
Liu, Qiang, and Yihao Feng. "Two methods for wild variational inference." arXiv preprint arXiv:1612.00081 (2016).

Song, Yang, et al. "Sliced score matching: A scalable approach to density and score estimation." Uncertainty in Artificial Intelligence. PMLR, 2020.

Song, Yang, and Stefano Ermon. "Generative modeling by estimating gradients of the data distribution." Advances in Neural Information Processing Systems. 2019.

---

### Official Review · AnonReviewer3 · 2020-10-28
**Very similar to a NeurIPS-19 paper**

**Rating:** 3
**Confidence:** 5

**Review:**

This paper adopted Stein's method to connect an explicit density estimator and an implicit sample generator to propose an objective function for deep generative learning.
This paper is written well and the organization is very clear.
However, this paper is very similar to a NeurIPS-19 paper (Two Generator Game: Learning to Sample via Linear Goodness-of-Fit Test).
Besides, the authors didn't cite this paper in the current submission.

First, the top-level idea is the same.
1. They all adopted two generative models. One is explicit and the other is implicit.
2. They all adopted Stein's method to connect these two generative models.

Second, important technical details are similar.
1. They all used energy-based model in the explicit part.
The energy-based model is used to mimic the underlying distribution of the real data.
2. They all used Stein's method to avoid solving the normalization constant.
Stein's method is a likelihood-free method that depends on the distribution only through logarithmic derivatives.
When taking derivatives, the normalization constant will be eliminated.

Third, these two papers have the same target.
They hope the explicit generative model characterize the formulation of the distribution and
the implicit one produces vivid or genuine-looking images.

The novel part of this submission is the introduction of kernel Sobolev dual norm and Moreau-Yosida regularization.

There is a big gap between the optimization formulations in Equation (3), Theorem 1 and Theorem 2 and the experimental results shown in Section 5.
Besides, there are no open source codes provided.
It is very hard for me to figure out the details of the experiments and meantime to check the reproducibility of this paper.

In summary, I hope that the authors will correctly cite the closely related NeurIPS-19 paper,
and clearly demonstrate their completely new contributions as compared to the NeurIPS-19 paper.

Since ICLR is a highly selective conference,
the originality and significance of one submission will always be in the first priority.
Although the writing quality of this paper is good, I cannot accept this paper in current state.

---

### Official Review · AnonReviewer1 · 2020-10-29
**The paper gives a solid and comprehensive analysis and implementation of the idea of Stein bridge, but the motivation may need further discussion.**

**Rating:** 6
**Confidence:** 4

**Review:**

Pros:
* The idea to bind an explicit and implicit generative model and study the effect on both models is a valid research topic.
* The method seems novel and inspiring, and the paper also shows a theoretical understanding of the proposed method, which is technically nontrivial.
* The presentation is clear and pleasing (e.g., content organization, background, Fig. 1). The paper also includes a detailed review on existing works.
* The paper presents comprehensive experiments, and the results are promising.

Cons:
* On the motivation.
  - Would it be too costly to train two models for one task just to alleviate the problem of one model? It seems to hide the problems of each model and serves as a black-box solution. The theoretical analysis makes things better, but the explanations may seem to be like "side effects" but not a direct solution targeting on the problems. Moreover, both the explicit and implicit models have the same amount of knowledge from data: one model cannot provide more information to the other model beyond the training dataset. How to understand the improvement under this perspective?
  - For training an explicit model, the mode-collapse behavior may be due to the usage of the Stein discrepancy. Training by maximizing likelihood (i.e., minimizing forward KL divergence) via classical methods e.g. contrastive divergence may already circumvent this problem.
* On the theory. It may be better to explain why "By smoothing the Stein critic, the Stein bridge encourages the energy model
to seek more modes in data instead of focusing on some dominated modes". How does it make the Stein discrepancy more picky on missing a mode?
* On the experiments.
  - I see in the supplement that a hyperparameter searching is conducted, but I did not find the metric to select them. Is it done by AUC / IS / FID / MMD / HSR / manual visualization evaluation? Results of "WGAN + something" may be sensitive to hyperparameters and maybe they should not be worse than the vanilla WGAN (taking zero regularization).
  - In Figs. 4 and 5, how are the density estimation of implicit models GAN/WGAN and samples of explicit models DEM/DGM got visualized? Do they rely on techniques like kernel density estimation or MCMC? If yes, how to make sure these techniques do not affect the outcome?
  - The definition of High-quality Sample Rate (HSR) may value mode collapse, since it gives a high score if all generated samples are on the center of one mode. For the result in Fig. 8(a), may be the HSR needs to drop to be faithful to data.
  - The experiment is comprehensive and shows the desired improvements. But maybe a comparison with other explicit model training methods (contrastive divergence, annealed Langevin dynamics, etc.) that avoid mode-collapse is also needed.

---

### Official Review · AnonReviewer4 · 2020-10-29
**Stein bridge is proposed to facilitate training both implicit and explicit models**

**Rating:** 5
**Confidence:** 3

**Review:**

In this paper, the task is to train an implicit and an explicit model simultaneously via GAN setting and a new regularizer called "stein bridge", which is constructed from the kernel Stein discrepancy between the implicit and explicit models. The idea of adding such regularization, with the notion of mutual regularization of two models, is interesting. The proposed regularization term is clearly presented, the illustration of stablizing the training procedure, and the empirical results are clearly shown and discussed. The sample quality from the generative models are compared.

There are some parts that remain unclear or can be further emphasized.
 It is said that training both explicit and implicit densities are more helpful to the whole procedure. Despite the cited literature reviews, it is unclear to me, in this paper presentation, why is this so?
In the paper, the implicit network is parameterized by \theta as G_{\theta} while the explicit EBM is parametrized by \phi, as p_{\phi}.
Before the Stein bridge is introduced:
How do \theta and \phi interact? From figure1 it seems they do not interact during training but only coupled via the objective. In addition, which of the model (implicit or explicit) is used as the final outcome?
After the Stein bridge is introduced:
stein bridge tries to minimize the Stein discrepancy between implicit and explict models. How is the EBM chosen so that the density class is rich enough? How are \lambda_1 and \lambda_2 chosen to balance three terms?

How does the training of generative model compared to the learning procedure in "Deep energy estimator networks." Saremi, Saeed, et al.  2018, which learns a generative model from score-matching based criterion?

Thanks for the presentation.

---

### Public Comment · ~Jianwen_Xie1 · 2020-11-14
**missing related works about EBMs**

Dear Authors and Reviewers,

We found that the current paper missed some important references about pioneering works that are related to energy-based generative models parameterized with deep net energy.

The first paper that proposes to train an energy-based model parameterized by modern deep neural network and learned it by Langevin based MLE is in (Xie. ICML 2016) [1]. The model is called generative ConvNet, because it can be derived from the discriminative ConvNet. This is also the first paper to formulate modern ConvNet-parametrized EBM as exponential tilting of a reference distribution, and connect it to discriminative ConvNet classifier. That is, EBM is a generative version of a discriminator. (Xie. ICML 2016) [1] originally studied such an EBM model on image generation theoretically and practically in 2016.

(Xie. CVPR 2017) [2] (Xie. PAMI 2019) [3] proposed to use Spatial-Temporal ConvNet as the energy function in EBMs for video generation. The model is called Spatial-Temporal generative ConvNet.

(Xie. CVPR 2018) [4] also proposed to use volumetric 3D ConvNet as the energy function for 3D shape pattern generation. It is called 3D descriptor Net.

Also, the Generative Cooperative Nets (CoopNets) (Xie. PAMI 2018)[5] and (Xie. AAAI 2018) [6], which jointly trains an EBM and a generator network by MCMC teaching.

Those are the more original and earlier papers for deep EBMs with ConvNet as energy function than what you have cited, e.g., [7](Yilun Du and Igor Mordatch, 2019).

References:

[1] A Theory of Generative ConvNet. Jianwen Xie *, Yang Lu *, Song-Chun Zhu, Ying Nian Wu (ICML 2016)

[2] Synthesizing Dynamic Pattern by Spatial-Temporal Generative ConvNet Jianwen Xie, Song-Chun Zhu, Ying Nian Wu (CVPR 2017)

[3] Learning Energy-based Spatial-Temporal Generative ConvNet for Dynamic Patterns Jianwen Xie, Song-Chun Zhu, Ying Nian Wu IEEE Transactions on Pattern Analysis and Machine Intelligence (TPAMI) 2019

[4] Learning Descriptor Networks for 3D Shape Synthesis and Analysis Jianwen Xie *, Zilong Zheng *, Ruiqi Gao, Wenguan Wang, Song-Chun Zhu, Ying Nian Wu (CVPR) 2018

[5] Cooperative Training of Descriptor and Generator Networks. Jianwen Xie, Yang Lu, Ruiqi Gao, Song-Chun Zhu, Ying Nian Wu. IEEE Transactions on Pattern Analysis and Machine Intelligence (TPAMI) 2018

[6] Cooperative Learning of Energy-Based Model and Latent Variable Model via MCMC Teaching. Jianwen Xie, Yang Lu, Ruiqi Gao, Ying Nian Wu. AAAI 2018.

[7] Yilun Du and Igor Mordatch. Implicit generation and modeling with energy based models. In Advances in Neural Information Processing Systems, pages 3603–3613, 2019

Thank you!

---

### Decision · Program_Chairs · 2021-01-07
**Final Decision**

**Decision:**

Reject

**Comment:**

This paper presents "stein bridge", a joint training framework that connects an explicit (unnormalized) density estimator and an implicit sample generator via Stein discrepancy. The idea and methodology are valid and of interest. But the raised concerns were not properly addressed.